# Noise-corrected GRPO: From Noisy Rewards to Unbiased Gradients

Omar El Mansouri [1]    Fathinah Asma Izzati [2]    Mohamed El Amine Seddik [3]    Salem Lahlou [1]

## Abstract

Reinforcement learning from human feedback (RLHF) or verifiable rewards (RLVR), the standard paradigm for aligning LLMs or building recent SOTA reasoning models, is highly sensitive to noise from inconsistent or erroneous rewards. Yet, the interaction between such noise and widely used group-based policy optimization methods remains underexplored. We introduce a noise-robust Group Relative Policy Optimization (GRPO) and Done Right GRPO (Dr.GRPO) framework that explicitly models reward corruption as Bernoulli noise. Our method applies noise correction after estimating reward flip probabilities to debias the learning signal, yielding unbiased gradient estimates. Theoretical analysis shows that group-based methods inherently mitigate individual-level noise, and our correction strategy amplifies this robustness. Empirically, we observe consistent improvements across math and code tasks when applying our noise correction to standard reward model usage, with particular gains of up to 6.7 percentage points in accuracy on math tasks and 1.5 on code tasks under realistic reward model conditions. This work bridges label-noise correction from supervised learning with modern RLHF, offering both theoretical insights and a practical algorithm for noisy real-world deployment.

## 1. Introduction

The evolution of Large Language Model (LLM) alignment has been driven by the search for more precise optimization signals. While Reinforcement Learning from Human Feedback (RLHF) (Ouyang et al., 2022) succeeded in cap-

[1]Department of Machine Learning, Mohamed bin Zayed University of Artificial Intelligence, Abu Dhabi, UAE [2]Department of Robotics, Khalifa University, Abu Dhabi, UAE [3]Technology Innovation Institute, Abu Dhabi, UAE. Correspondence to: Omar El Mansouri <Omar.ElMansouri@mbzuai.ac.ae>.

*Proceedings of the 43rd International Conference on Machine Learning*, Seoul, South Korea. PMLR 306, 2026. Copyright 2026 by the author(s).

turing subjective linguistic norms, it shows limitations in rigorous domains like mathematics and programming. In these fields, "correctness" is not a matter of opinion but a verifiable fact. This shift has catalyzed the adoption of Reinforcement Learning with Verifiable Rewards (RLVR) (Lambert et al., 2024), which replaces human judgment with automated, objective signals, such as code execution or gold-answer matching, to drive scalable and reproducible optimization.

However, the move toward automation introduces a new problem: the verifier itself. In practice, the reward signals we rely on are flawed, creating a "corrupted" feedback loop. For example LLM as a judge often exhibit systematic biases; they can be tricked into "False Positives" (accepting incorrect solutions) simply by the presence of persuasive phrases like "Let's solve this step by step." (Zhao et al., 2025). Moreover, exact-match criteria or regex-based parsers often suffer from "False Negatives", failing to recognize mathematically equivalent answers (equivalent fractional representations for example) or variations in text formatting (Li et al., 2025b). This noise stems from several unavoidable sources, such as human labeling inconsistencies in the training (Zhang et al., 2024), reward model approximation errors (Christian et al., 2025), distributional shift (LeVine et al., 2023), and adversarial manipulations (Bukharin et al., 2025)

In this work, we formally model these imperfections by **conceptualizing the observed reward as a noisy observation of a true, latent reward**. We posit the existence of a deterministic, ground-truth reward function $r^*(q, o)$ that perfectly reflects human preference for a response $o$ given a prompt $q$. The reward model $r_\phi(q, o)$, in practice, does not have access to $r^*$. Instead, it observes a corrupted version of it through a noisy channel. Let the observed reward $\tilde{r}$ be given by: $\tilde{r}(q, o) = r_\phi(q, o) = \mathcal{C}(r^*(q, o))$, where $\mathcal{C}$ is a stochastic noise channel that corrupts the true reward signal. Since RLVR setting involves binary rewards, i.e defined as a deterministic function $r \colon Q \times O \to \{0, 1\}$ where $Q$ denotes the space of queries and $O$ the space of responses, $\mathcal{C}$ will be modelled as Bernoulli query dependent.

This formulation starkly illustrates the fundamental challenge of RLVR: we perform reinforcement learning to optimize the expectation of the true reward $\mathbb{E}[r^*]$:

$$\pi^* = \arg\max_\pi \mathbb{E}_{o \sim \pi(\cdot|q)}\big[r^*(q, o)\big],$$

but our optimization process is merely guided by the noisy surrogate $\tilde{r}$. Directly applying policy optimization algorithms like GRPO (Shao et al., 2024) or Dr.GRPO (Liu et al., 2025) to $\tilde{r}$ without accounting for the noise channel $\mathcal{C}$ can lead to performance degradation: we prove in Section 4 that noise attenuates the advantage signal and yields convergence to a strictly worse fixed point.

To bridge this gap, we propose **a novel denoising framework for policy optimization**, we first propose a rigourous theoretical analysis of the noise effect, then by estimating the parameters of the noise, we apply a correction mechanism inspired by techniques from learning with noisy labels in the context of supervised learning, to recover a debiased estimate of the true advantage signal before each policy update. We integrate this approach into the GRPO framework and its variant Dr.GRPO, enhancing its robustness and enabling more effective alignment under realistic, noisy reward conditions.

**Our contributions are the following:**

- **A formal study of GRPO and Dr.GRPO under noisy rewards** where we prove that reward corruption attenuates the learning signal and leads to convergence to a strictly worse policy. We also analyze the gradient dynamics under noise.

- **Principled noise correction with closed-form update and guarantee.** Adapting label-noise theory, we propose a practical denoising algorithm that estimates the noise corruption and applies a correction to the policy gradient update, making it robust to said noise, and we derive a monotonic-improvement bound. We also provide an explicit sample-complexity bound for the noise estimation.

- **Theory-driven empirical validation** on math and code generation tasks where we follow a two-stage experimental protocol:

  - **Perfect Assumption Validity Stage:** We first simulate a noisy channel applied to our ground truths (math answers/code execution) with fixed, known class-conditional flip rates to enforce perfect assumption validity. This allows us to verify that the empirical behavior of our method aligns with our theoretical predictions. Under synthetic noise conditions, our method not only recovers from performance degradation but in some cases (e.g., Llama-3.2-1B) even exceeds the noiseless baseline performance, suggesting additional regularization benefits.

  - **Realistic Estimation Stage:** In a more practical setting, we take an existing pre-trained reward model, estimate its flip rates directly from data, and then apply our correction method using these estimated values. Specifically, we achieve accuracy improvements of up to 6.7 percentage points compared to using the same reward models without correction for math tasks and 1.5 percentage points for code tasks .

> **Note**
>
> To maintain clarity and readability in the main text, all proofs of theorems, propositions, and corollaries are provided in Appendix A.

## 2. Background

This section reviews the policy optimization algorithms (GRPO, and its modern variant Dr.GRPO) that form the foundation of our work.

**Group-Relative Policy Optimisation:** Unlike PPO (Schulman et al., 2017), which relies on a learned critic and GAE for advantage estimates combined with ratio clipping for stability, GRPO (Shao et al., 2024) dispenses with critics and instead normalises rewards within each prompt. Defining $p_k(q) = \mathbb{E}_{o \sim \pi_k}[r(q, o)]$ and $\sigma_k(q) = \sqrt{Var_{o \sim \pi_k}[r(q, o)]}$, the centred, scaled advantage is $A_k(q, o) = (r(q, o) - p_k(q))/\sigma_k(q)$, GRPO maximises $\mathcal{L}_{\text{GRPO}}^{\pi_k}(\pi) = \mathbb{E}_{q \sim \rho_Q, o \sim \pi}[A_k(q, o)] - \beta \, \text{KL}(\pi \| \pi_{\text{ref}})$, which cuts variance. Indeed, dividing by the per-prompt standard deviation normalises advantage magnitudes so high-variance prompts contribute smaller advantages while consistent prompts with low variance contribute larger ones, lowering gradient variance and making training more stable. However, dividing by $\sigma_k(q)$ introduces a difficulty bias as we will see later.

**Done Right GRPO:** Dr.GRPO (Liu et al., 2025) keeps the group-mean baseline but drops the variance and length divisors, i.e using $A_k^{\text{Dr}}(q, o) = r(q, o) - p_k(q)$. This unbiased surrogate stabilises chain-of-thought length and improves sample efficiency, but a symmetric clip still hampers exploration, and identical group rewards yield zero gradient. This approach will appear as a special case of our upcoming generalization.

**Positioning within prior work.** Several lines of work have sought to enhance robustness in RLHF, but primarily focus on the reward modeling stage. For example, Reward-Robust RLHF (Yan et al., 2024) employs Bayesian reward model ensembles to characterize uncertainty, while (Bukharin et al., 2024) formulates reward learning as an $\ell_1$-regularized problem to downweight sparse outliers. Other methods, such as contrastive reward approaches (Shen et al.), enhance robustness by penalizing reward uncertainty but

lack formal noise modeling, and REINFORCE++ (Hu et al.) improves advantage estimation stability but remains vulnerable to inherent reward noise. Crucially, these methods address robustness upstream; the interaction between reward noise and group-based policy optimization remains underexplored, leaving a need for a principled correction mechanism at the policy update level.

## 3. Preliminaries on the Noiseless Case

This section formally defines the GRPO objective and advantage function in the ideal, noiseless setting. We then present key theoretical guarantees (Theorems 3.2 and 3.3) that characterize its convergence and improvement properties. Section 4 extends these results to the noisy case.

We begin by recalling the core objective and update rule of GRPO in the noiseless setting. GRPO is an iterative optimization process where, at each iteration $k$, a policy $\pi_k$ is updated to maximize a surrogate objective, and we denote $\pi_{\text{ref}}$ the base model policy at iteration $k = 0$ (before the training). The surrogate loss is defined as the expected advantage for a new policy $\pi$ using samples from the current policy $\pi_k$ :

$$
\begin{aligned}
L_{\pi_k}\big(\pi(\cdot \mid q)\big) &= \mathbb{E}_{o \sim \pi_k(\cdot \mid q)}\left[\frac{\pi(o \mid q)}{\pi_k(o \mid q)} A_{\pi_k}(o, q)\right] \\
&= \mathbb{E}_{o \sim \pi(\cdot \mid q)}\left[\frac{r^*(q, o) - \mathbb{E}_{o' \sim \pi_k} r^*(q, o')}{\sigma_k(q)}\right].
\end{aligned}
$$

The transition from the first to the second line is due to importance sampling, and the transition from the second to the third line is due to the GRPO advantage as defined in Section 2.

**Definition 3.1** (success probability). For a given prompt $q$ and policy $\pi$, we define the *success probability* as the expected reward :

$$
p_\pi(q) = \mathbb{E}_{o \sim \pi(\cdot \mid q)}[r^*(q, o)].
$$

This represents the current probability of generating a correct response, i.e the capacity of the LLM (which follows the policy $\pi$) to generate correct outputs. The variance of the reward is :

$$
\sigma_\pi(q)^2 = \mathbb{E}_{o \sim \pi(\cdot \mid q)}\big[(r^*(q, o) - p_\pi(q))^2\big] = p_\pi(q)\big(1 - p_\pi(q)\big)
$$

since $r^*(q, o) \sim \text{Bernoulli}\big(p_\pi(q)\big)$.

A crucial guarantee in RL is that an algorithm should improve the policy at each step. The following theorem from Mroueh et al. (2025), provides a lower bound on the actual improvement $p_{\pi_{k+1}}(q) - p_{\pi_k}(q)$ achieved by maximizing the surrogate loss.

**Theorem 3.2** (Policy Improvement). *For any policy $\pi$,*

$$
\mathbb{E}_{q \sim \rho_Q}\big[p_\pi(q) - p_{\pi_k}(q)\big] \geq \mathbb{E}_{q \sim \rho_Q}\big[L_{\pi_k}(\pi(\cdot \mid q))\big]
$$
$$
- 2\sqrt{\mathbb{E}_{q \sim \rho_Q}\left(\frac{1 - \sigma_k(q)}{\sigma_k(q)}\right)^2}\sqrt{\mathbb{E}_{q \sim \rho_Q}\text{TV}^2(\pi(\cdot \mid q)\|\pi_k(\cdot \mid q))}.
$$
(1)

Maximizing the right-hand term at each iteration leads to a monotonic improvement (Mroueh et al., 2025).

Interestingly, the penalty term in Theorem 3.2 highlights a potential issue with the GRPO formulation: its dependence on $\frac{1 - \sigma_k(q)}{\sigma_k(q)}$ as discussed in DAPO (Yu et al., 2025) and DISCO (Li et al., 2025a). This term introduces a difficulty bias as illustrated in Figure 1.

To further understand the dynamics of GRPO, we analyze the optimal solution to the constrained policy optimization problem. We fix a $\beta > 0$ and, at each iteration, we have:

$$
\pi_{k+1} = \max_\pi \mathbb{E}_{q \sim \rho_Q}\big[L_{\pi_k}(\pi(\cdot \mid q)) - \beta \, \text{KL}(\pi(\cdot \mid q)\|\pi_{\text{ref}}(\cdot \mid q))\big].
$$
(2)

The link between TV/KL is given by Pinsker's inequality.

Theorem 3.3 shows that the success probability $p_{\pi_k}(q)$ follows a deterministic recursion relation:

**Theorem 3.3** (Recursion for success probability (Mroueh, 2025)). *Denoting $p_{\pi_k}(q) = \mathbb{E}_{o \sim \pi_k(\cdot \mid q)}[r^*(q, o)]$, probability success at iteration $k$, we have:*

$$
p_{\pi_k}(q) = h_{\varepsilon, ref}(p_{\pi_{k-1}}(q)),
$$
$$
where \quad h_{\varepsilon, ref}(p) = \frac{1}{1 + \frac{1 - p_{ref}}{p_{ref}} \exp\left(-\frac{1}{\beta\sqrt{p(1-p) + \varepsilon}}\right)}.
$$

From this recursion, we can also derive the following key property of GRPO: In the noiseless case, it is guaranteed to improve upon the reference policy $\pi_{ref}$ at every step:

**Proposition 3.4** (Monotonic improvement). *For all $k$, the following inequality holds: $p_{\pi_k}(q) > p_{\text{ref}}$.*

In summary, the noiseless GRPO algorithm possesses strong theoretical guarantees, including a closed-form update and monotonic improvement over the reference policy. Next, we will see how these properties are affected by reward noise.

## 4. Noisy GRPO

In this section, we introduce a model for reward corruption. We consider a setting where we do not observe the true binary reward $r^*(q, o) \in \{0, 1\}$ directly but a corrupted version $\tilde{r}(q, o)$, generated via a corruption channel $\mathcal{C}(q)$ that depends on the query $q$. We analyze its effect on the GRPO objective, derive a new policy improvement lower bound, and show that noise systematically attenuates the learning signal and leads to convergence to a worse fixed point.

**Binary corruption model.** Since we are working with binary rewards, we adopt a Bernoulli noise model. This captures the core phenomenon of labels being flipped with certain probabilities defined here :

$$
\Pr(\tilde{r} \mid r^*(q,o)) = \begin{cases} \rho^-(q), & \tilde{r} = 0,\ r^*(q,o) = 1, \\ 1 - \rho^-(q), & \tilde{r} = 1,\ r^*(q,o) = 1, \\ 1 - \rho^+(q), & \tilde{r} = 0,\ r^*(q,o) = 0, \\ \rho^+(q), & \tilde{r} = 1,\ r^*(q,o) = 0. \end{cases}
$$

$\rho^+(q)$ and $\rho^-(q)$ being, respectively, the "False Positive rate" and the "False Negative rate".

**Stochastic representation:** We equivalently write $\tilde{r}$ as a function of an auxiliary random variable $\xi \sim U[0,1]$ with $U[0,1]$, being the uniform law on $[0,1]$, and $\xi$ is independent from $(q,o)$ :

$$
\tilde{r}(q,o,\xi) = \begin{cases} \mathbf{1}_{\{\xi \leq \rho^+(q)\}} & \text{if } r^*(q,o) = 0, \\ \mathbf{1}_{\{\xi \leq 1-\rho^-(q)\}} & \text{if } r^*(q,o) = 1. \end{cases} \tag{3}
$$

Here, $\mathbf{1}_{\{\cdot\}}$ denotes the indicator function, which returns 1 if the condition is true and 0 otherwise.

Equivalently :

$$
\tilde{r}(q,o,\xi) = (1 - r^*(q,o))\,\mathbf{1}_{\{\xi \leq \rho^+(q)\}} + r^*(q,o)\,\mathbf{1}_{\{\xi \leq 1-\rho^-(q)\}}
$$

**Expectation:** To understand the bias introduced by the noise, we compute the expectation of the noisy reward over the joint law $\xi$ and $o \sim \pi_k(\cdot \mid q)$ :

$$
\mathbb{E}_\xi \mathbb{E}_o[\tilde{r}(q,o,\xi)] = \rho^+(q) + (1 - \rho^+(q) - \rho^-(q))\,\mathbb{E}_o[r^*(q,o)]
$$
$$
=: \mu_{\pi_k}(q). \tag{4}
$$

**Variance:** Similarly, the variance of the noisy reward captures the combined uncertainty from the policy's performance and the label flipping: The variance under the joint law $\xi$ and $o \sim \pi_k(\cdot \mid q)$ is :

$$
\mathrm{Var}_{o \sim \pi_k, \xi}[\tilde{r}(q,o,\xi)] = \mu_{\pi_k}(q)\,(1 - \mu_{\pi_k}(q)) := \tilde{\sigma}_k(q)^2.
$$

The expectation $\mu_{\pi_k}(q)$ is a biased version of the true mean $p_{\pi_k}(q)$. The variance $\tilde{\sigma}_k(q)^2$ now captures uncertainty from both the policy and the noise process.

**Centered and normalized objective.** In the noisy case we do not observe $L_{\pi_k}(\pi(\cdot \mid q))$ but its corrupted counterpart $\tilde{L}_{\pi_k}(\pi(\cdot \mid q))$:

$$
\tilde{L}_{\pi_k}(\pi(\cdot \mid q)) = \mathbb{E}_{\xi,\,o \sim \pi(\cdot \mid q)}\left[\tilde{A}(q,o,\xi)\right]
$$
$$
= \frac{\mathbb{E}_{\xi,\,o \sim \pi(\cdot \mid q)}[\tilde{r}(q,o,\xi)] - \mathbb{E}_{\xi,\,o \sim \pi_k}[\tilde{r}(q,o,\xi)]}{\sqrt{\mathrm{Var}_{\xi,\,o \sim \pi_k}[\tilde{r}(q,o,\xi)]}}.
$$

Taking expectation over $\xi$ yields:

$$
\tilde{L}_{\pi_k}(\pi(\cdot \mid q)) = \frac{(1 - \rho^+(q) - \rho^-(q))\left(\mathbb{E}_{o \sim \pi}[r^*(q,o)] - \mathbb{E}_{o \sim \pi_k}[r^*(q,o)]\right)}{\tilde{\sigma}_k(q)} \tag{5}
$$

Mirroring our analysis in the noiseless case, we can derive a policy improvement lower bound for the noisy setting.

**Theorem 4.1** (Policy Improvement in the noisy case). *For any policy $\pi$,*

$$
\mathbb{E}_{q \sim \rho_Q}[p_\pi(q) - p_{\pi_k}(q)] \geq \mathbb{E}_{q \sim \rho_Q}[\tilde{L}_{\pi_k}(\pi(\cdot \mid q))]
$$
$$
- 2\sqrt{\mathbb{E}_{q \sim \rho_Q}\left(\frac{1 - \rho^+(q) - \rho^-(q) - \tilde{\sigma}_k(q)}{\tilde{\sigma}_k(q)}\right)^2}\sqrt{\mathbb{E}_{q \sim \rho_Q}\mathrm{TV}^2(\pi(\cdot \mid q)\|\pi_k(\cdot \mid q))}
$$

**Improvement in the noisy case:** The right-hand term of the inequality in Theorem 4.1 valuated at $\pi = \pi_k$ is zero (because $\tilde{L}_{\pi_k}(\pi_k(\cdot \mid q)) = 0$ and $\mathrm{TV}(\pi_k(\cdot \mid q)\|\pi_k(\cdot \mid q)) = 0$), so if we maximize this term over the set of all policies, it is necessarily positive when evaluated for the optimal policy denoted $\pi_{k+1}$, ensuring that $p_{\pi_{k+1}}(q) \geq p_{\pi_k}(q)$. However, in practice we don't maximise this exact right term. Thus, this theorem guarantees that maximizing the surrogate loss $\tilde{L}_{\pi_k}(\pi)$ leads to a policy $\pi_{k+1}$ that improves the true objective $p_\pi(q)$, provided the penalty term (which depends on the TV divergence between policies and the inverse standard deviation) is not too large. It formalizes the trade-off between improvement and stability that is also visible in the noiseless case.

This improvement guarantee is a crucial property, ensuring that the optimization process does not diverge. However, while the algorithm is guaranteed to improve, the quality and efficiency of this improvement are significantly impacted by reward noise. We now analyze how noise degrades the learning signal and alters the difficulty bias observed in the noiseless case.

**Difficulty bias noisy case:** Since $\frac{1 - \rho^+(q) - \rho^-(q)}{\tilde{\sigma}_k(q)} \leq \frac{1}{\sigma_k(q)}$, it follows that $\mathbb{E}_q[\tilde{L}_{\pi_k}(\pi)] \leq \mathbb{E}_q[L_{\pi_k}(\pi)]$. This confirms the intuitive fact that noise attenuates the advantage signal.

The structure of the bound is similar to Theorem 3.2 but now includes the factor $(1 - \rho^+ - \rho^-)$ inside the surrogate loss. This reflects the degradation of the learning signal caused by the noise. This inequality confirms the intuitive fact that noise attenuates the advantage signal. However, an interesting trade-off emerges. While the signal is dampened, the penalty term in the new lower bound (Theorem 4.2) may be better behaved than in the noiseless case (Theorem 3.2). This 'flattening' effect is illustrated in Figure 1. But, since we fix the regularization parameter $\beta$ in the existing literature, we do not benefit from this flattening effect. We believe future works can benefit from this observation.

To precisely characterize the impact of noise on convergence, we derive the recurrence relation for the success

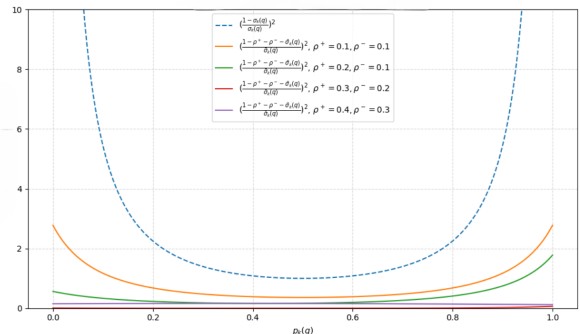

*Figure 1.* Effect of Label Noise on the Lower Bound. Noise flattens the penalization term. Specifically, the term $\frac{1-\rho^+ + \rho^- - \tilde{\sigma}_k(q)}{\tilde{\sigma}_k(q)}$ might not explode as severely as $\frac{1-\sigma_k(q)}{\sigma_k(q)}$ when $p_k(q)$ is near 0 or 1, because $\tilde{\sigma}_k(q)$ is bounded away from zero by the noise.

probability under the noisy update.

**Theorem 4.2** (Recursion in the noisy case)**.** *Denoting* $p_{\pi_k}(q) = \mathbb{E}_{o \sim \pi_k(\cdot|q)}(r^*(q,o))$, *the probability of success at iteration k. Under the assumption that* $1 - \rho^+(q) - \rho^-(q) > 0$, *we have:*

$$p_{\pi_k}(q) = \tilde{h}_{\varepsilon,ref}(p_{\pi_{k-1}}(q)),$$

*where* $\quad \tilde{h}_{\varepsilon,ref}(p) = \dfrac{1}{1 + \frac{1-p_{ref}}{p_{ref}} \exp\left(-\frac{1-\rho^+(q)-\rho^-(q)}{\beta\sqrt{F(p)(1-F(p))+\varepsilon}}\right)},$

*with* $F(p) = \rho^+(q) + (1 - \rho^+(q) - \rho^-(q)) \cdot p$. *Moreover,* $\forall p \in [0,1]$, $\tilde{h}_{\varepsilon,ref}(p) < h_{\varepsilon,ref}(p)$.

**Corollary 4.3** (Noise degrades performance at equilibrium)**.** *Denoting* $p^*_{noisy}$ *and* $p^*_{noiseless}$ *the fixed points of* $\tilde{h}_{\varepsilon,ref}$ *and* $h_{\varepsilon,ref}$ *respectively, we have:*

$$p^*_{noisy} < p^*_{noiseless}.$$

This implies that the fixed point performance, the final accuracy the policy converges to, is strictly lower when rewards are corrupted. This formally quantifies the performance cost of reward noise.

**Proposition 4.4** (Amplification in the noisy case)**.** *For all* $k$, *the following inequality holds in the noisy case too:* $p_{\pi_k}(q) > p_{ref}$.

Proposition 4.4 shows that despite noise, the fundamental property of improvement over the reference policy is retained. This is important for ensuring the algorithm remains valid.

Section 4 reveals a somewhat paradoxical picture: GRPO and Dr.GRPO retain improvement over the reference policy even under flip noise (Proposition 4.4), yet Corollary 4.3 establishes that the recursion converges to a *strictly worse*

fixed point, i.e., $p^*_{noisy} < p^*_{noiseless}$. Stability alone is therefore insufficient; the attenuated learning signal prevents recovery of the clean optimum regardless of training duration. This motivates the central question of the next section: can we design an update whose gradient matches the clean-reward objective while preserving the simplicity of group-based optimization?

## 5. Noise Correction

### 5.1. Noise debiasing

A first instinct is to borrow tools from learning with noisy labels (Natarajan et al., 2013). They show that if the class-conditional noise rates are known, one can transform any noisy loss function into an unbiased estimator of the true, clean loss function. This allows standard optimization algorithms to converge to the correct solution as if they were trained on clean data.

We extend their correction to the RLVR context. The key insight is that the noisy reward $\tilde{r}$ can be transformed into a new random variable $\hat{r}$ whose expectation is equal to the true, unobserved reward $r^*$, removing the bias introduced by the noise channel, provided the true noise rates $\rho^+$ and $\rho^-$ are known:

$$\hat{r}(q,o,\xi) = \begin{cases} \dfrac{1-\rho^+}{1-\rho^+-\rho^-}, & \text{if } \tilde{r}(q,o,\xi) = 1, \\[2mm] \dfrac{-\rho^+}{1-\rho^+-\rho^-}, & \text{if } \tilde{r}(q,o,\xi) = 0. \end{cases}$$

**Proposition 5.1** (Debiasing)**.** *$\hat{r}$ is an unbiased estimator of the true reward function* $r^*$:

$$\mathbb{E}_{\xi}\big[\hat{r}(q,o,\xi)\big] = r^*(q,o) \tag{6}$$

While tempting to use unbiased estimators, GRPO's standardization invariance renders linear reparameterizations $\hat{r} = a\tilde{r} + b$ ineffective. Because $\hat{A} = \text{sgn}(a)\tilde{A}$, linear shifts are canceled and scales are either preserved or flipped. This necessitates a more robust objective than simple linear correction to address systematic reward corruption.

> The failure of linear reparameterization reveals that the problem is the *interaction* between the biased mean and mismatched variance normalization. Our strategy: (i) apply the Natarajan correction to debias the *mean* reward signal, and (ii) choose $M_k(q)$ to recover the correct gradient *scale*. Setting $M_k(q) = \sigma_k(q)$ exactly recovers the noiseless update; since $\sigma_k(q)$ is unobserved, we estimate it via Proposition 5.3.

**Generalized objective.** To overcome this limitation while still benefiting from the debiasing's property, we propose a generalized objective where the standard deviation normalization $\sigma_k(q)$ is replaced by a more general, strictly positive

function $M_k(q)$:

$$L_{\pi_k}^{\text{Deb}}(\pi(\cdot \mid q)) = \frac{\mathbb{E}_{o\sim\pi}\left[r_\xi(q,o)\right] - \mathbb{E}_{o\sim\pi_k}\left[r_\xi(q,o)\right]}{M_k(q)},$$

where $r_\xi(q,o) := \mathbb{E}_\xi[\hat{r}(q,o,\xi)]$. This yields:

$$L_{\pi_k}^{\text{Deb}}(\pi(\cdot \mid q)) = \mathbb{E}_{o\sim\pi(\cdot\mid q)}\left[\frac{r^*(q,o) - \mathbb{E}_{o\sim\pi_k(\cdot\mid q)}\left[r^*(q,o)\right]}{M_k(q)}\right]$$

The following theorem provides a policy improvement guarantee for our generalized objective, where the tightness of the bound now depends on the choice of $M_k(q)$:

**Theorem 5.2.** *For any policy $\pi$,*

$$\mathbb{E}_{q\sim\rho_\mathcal{Q}}\left[p_\pi(q) - p_{\pi_k}(q)\right] \geq \mathbb{E}_{q\sim\rho_\mathcal{Q}}\left[L_{\pi_k}^{Deb}(\pi(\cdot \mid q))\right]$$
$$-2\sqrt{\mathbb{E}_{q\sim\rho_\mathcal{Q}}\left(\frac{1-M_k(q)}{M_k(q)}\right)^2}\sqrt{\mathbb{E}_{q\sim\rho_\mathcal{Q}}\text{TV}^2(\pi(\cdot \mid q)\|\pi_k(\cdot \mid q))}. \tag{7}$$

If we take $M_k(q) = 1$, we retrieve the Dr.GRPO (Liu et al., 2025) estimator.

If we take $M_k(q) = \sigma_k(q)$, we end up with exactly the same *lower bound* as in the *noiseless* case. Given that we do not have access to $\sigma_k(q)$, we propose to estimate it using the following proposition. The estimator works by calculating the variance of the debiased rewards and then subtracting the extra variance introduced by the stochastic correction process itself.

**Proposition 5.3** (Estimator of the true variance). *Let $q$ be fixed and observe $(o_i, \xi_i)_{i=1}^n$. Suppose we have access to $\rho^+(q)$ and $\rho^-(q)$, and define*

$$Z\big((o_i,\xi_i)_{i=1}^n\big) \equiv \widehat{\text{Var}}\big(\hat{r}(q,o,\xi)\big) - \frac{\bar{r}(q)\,\rho^-(q)(1-\rho^-(q))}{(1-\rho^+(q)-\rho^-(q))^2}$$
$$- \frac{(1-\bar{r}(q))\,\rho^+(q)(1-\rho^+(q))}{(1-\rho^+(q)-\rho^-(q))^2}, \tag{8}$$

*where $\bar{r}(q) \equiv \frac{1}{n}\sum_{i=1}^n \hat{r}(q,o_i,\xi_i)$ and $\widehat{\text{Var}}\big(\hat{r}(q,o,\xi)\big) \equiv \frac{1}{n-1}\sum_{i=1}^n\big(\hat{r}(q,o_i,\xi_i) - \bar{r}(q)\big)^2$. Then $Z$ is an unbiased and consistent estimator of the true variance $\text{Var}\big(r^*(q,o)\big)$.*

Theorem 5.4 formalizes the intuition from Section 4: under noisy rewards the GRPO gradient is multiplicatively shrunk by $(1 - \rho^+ - \rho^-)$ and the variance mismatch between clean and noisy rewards. This makes clear what correction must do: recover the clean mean signal and align the normalization so the gradient scale matches the noiseless update.

**Theorem 5.4** (Gradient analysis). *Let $\pi_\theta(\cdot \mid q)$ denote the LLM policy parameterized by weights $\theta$, and let $\pi_k$ be the current policy.*

*Then $\nabla_\theta \tilde{L}_{\pi_k}\big(\pi_\theta(\cdot \mid q)\big) = \alpha_k(q)\,\nabla_\theta L_{\pi_k}\big(\pi_\theta(\cdot \mid q)\big)$, where $\alpha_k(q) := (1 - \rho^+ - \rho^-)\frac{\sigma_k(q)}{\tilde{\sigma}_k(q)}$. Moreover, if we consider the (centered) Dr.GRPO reward-part surrogate*

$$L_{\pi_k}^{\text{Dr}}\big(\pi_\theta(\cdot \mid q)\big) := \mathbb{E}_{o\sim\pi_\theta(\cdot\mid q)}\left[r^*(q,o) - p_k(q)\right],$$

*and its noisy counterpart*

$$\tilde{L}_{\pi_k}^{\text{Dr}}\big(\pi_\theta(\cdot \mid q)\big) := \mathbb{E}_{\xi,o\sim\pi_\theta(\cdot\mid q)}\left[\tilde{r}(q,o,\xi) - \mu_k(q)\right],$$

*then $\nabla_\theta\tilde{L}_{\pi_k}^{\text{Dr}}\big(\pi_\theta(\cdot \mid q)\big) = (1-\rho^+-\rho^-)\,\nabla_\theta L_{\pi_k}^{\text{Dr}}\big(\pi_\theta(\cdot \mid q)\big).$*

*Applying our debisasing for Dr.GRPO case ($M_k = 1$) gives:*

$$\nabla_\theta L_{\pi_k}^{\text{Deb}}\big(\pi_\theta(\cdot \mid q)\big) = \nabla_\theta L_{\pi_k}^{\text{Dr}}\big(\pi_\theta(\cdot \mid q)\big)$$

*Applying it for GRPO case ($M_k = \sqrt{Z_k}$) gives :*

$$\nabla_\theta L_{\pi_k}^{\text{Deb}}\big(\pi_\theta(\cdot \mid q)\big) = \frac{\sigma_k(q)}{\sqrt{Z_k}}\nabla_\theta L_{\pi_k}\big(\pi_\theta(\cdot \mid q)\big)$$

For Dr. GRPO ($M_k = 1$), the correction yields an exactly unbiased gradient of the clean centered surrogate. For GRPO, exact recovery holds under the oracle choice $M_k(q) = \sigma_k(q)$. In our setup, we use $M_k(q) = \sqrt{Z_k(q)}$, (that we clip to handle negative estimates) yielding a consistent plug–in approximation whose multiplicative error concentrates around 1 as $Z_k$ concentrates.

---

**Algorithm 1** Noise-corrected GRPO/Dr.GRPO

---

**Input:** Training dataset $\mathcal{D}_{\text{train}}$, base policy $\pi_{\text{ref}}$, reward model $r_\phi$
**Output:** Optimized policy $\pi_\theta$
**Parameter:** $M = 1$ for Dr.GRPO, or $M = \sqrt{\max(Z,\epsilon)}$ with $\epsilon > 0$ for GRPO

**Stage 1: Estimate Noise Rates**
Sample $N$ queries (we took $N = 20\%$ of the dataset), corrupt half the correct answers (by modifying numbers in math tasks, adding code bugs in code tasks) in order to have balanced $\mathcal{D}_{\text{est}}$. Take $m_+ = m_- = N/2$
Compute noise rates on $\mathcal{D}_{\text{est}}$:
$\rho^+ \leftarrow \frac{1}{m_-}\sum \mathbf{1}[r_\phi = 1 \wedge r^* = 0]$ *(FPR)*
$\rho^- \leftarrow \frac{1}{m_+}\sum \mathbf{1}[r_\phi = 0 \wedge r^* = 1]$ *(FNR)*

**Stage 2: Robust Training**
**for** each training iteration on $\mathcal{D}_{\text{train}} \setminus \mathcal{D}_{\text{est}}$ **do**
    Sample batch $\mathbf{q} \sim \mathcal{D}$, generate $G$ responses per prompt: $\{o_1, \ldots, o_G\} \sim \pi_{\theta_{\text{old}}}$
    Get noisy rewards $\tilde{r}(q, o_i, \xi_i) = r_\phi(\mathbf{q}, o_i)$
    **Correction:** $\hat{r}_i \leftarrow \frac{\tilde{r}(q,o_i,\xi_i) - \rho^+}{1 - \rho^+ - \rho^-}$
    Compute advantage: $\hat{A}_i = \hat{r}_i - \frac{1}{G}\sum_{j=1}^G \hat{r}_j$
    Update $\pi_\theta$ via:
    $\mathcal{L}(\theta) = \mathbb{E}_{q\sim\mathcal{Q}}\left[\frac{1}{G}\sum_{i=1}^G \frac{\pi_\theta(o_i|q)}{\pi_{\theta_{\text{old}}}(o_i|q)}\frac{\hat{A}_i}{M}\right] - \beta\,\mathbb{E}_{q\sim\mathcal{Q}}\left[D_{\text{KL}}\big(\pi_\theta(\cdot \mid q) \,\|\, \pi_{\text{ref}}(\cdot \mid q)\big)\right]$
**end for**

---

### 5.2. Flip rates

So far we assumed access to the flip rates. In realistic RLHF/RLVR, these parameters are not given, and any practical

method must estimate them from data. We next quantify how estimation error propagates into the bound and the update, and provide a simple two-stage procedure used in our experiments.

Suppose we work with estimates $\hat{\rho}^+(q)$ and $\hat{\rho}^-(q)$. We thus work with

$$\hat{r}(q, o) = \frac{r_\phi(q, o) - \hat{\rho}^+(q)}{1 - \hat{\rho}^+(q) - \hat{\rho}^-(q)}, \tag{9}$$

instead of

$$\hat{r}(q, o) = \frac{r_\phi(q, o) - \rho^+(q)}{1 - \rho^+(q) - \rho^-(q)}. \tag{10}$$

**Theorem 5.5.** *Denoting* $M_k'(q) = \frac{1 - \hat{\rho}^+(q) - \hat{\rho}^-(q)}{1 - \rho^+(q) - \rho^-(q)} M_k(q)$, *the following inequality holds, with* $L_{\pi_k}^{\mathrm{Deb}}(\pi(\cdot \mid q))$ *computed with the estimated flip rates:*

$$\mathbb{E}_{q \sim \rho_Q}\big[p_\pi(q) - p_{\pi_k}(q)\big] \geq \mathbb{E}_{q \sim \rho_Q}\big[L_{\pi_k}^{\mathrm{Deb}}(\pi(\cdot \mid q))\big]$$
$$- 2\sqrt{\mathbb{E}_{q \sim \rho_Q}\left(\frac{1 - M_k'(q)}{M_k'(q)}\right)^2} \times \sqrt{\mathbb{E}_{q \sim \rho_Q}\mathrm{TV}^2\big(\pi(\cdot \mid q) \,\|\, \pi_k(\cdot \mid q)\big)}$$

This result shows that errors in estimating the noise rates translate to a scaling factor on the normalization function $M_k(q)$. While this introduces some bias, the fundamental correction structure remains. Finally, we provide an explicit quantitative sample-complexity result regarding the estimation-error propagation in Theorem 5.5.

**Proposition 5.6.** *Let* $\lambda := \dfrac{1 - \hat{\rho}^+ - \hat{\rho}^-}{1 - \rho^+ - \rho^-}$ *the estimation error rate. Let* $\rho^+ := \mathrm{Pr}(\tilde{r} = 1 \mid r^* = 0)$, $\rho^- := \mathrm{Pr}(\tilde{r} = 0 \mid r^* = 1)$, *Suppose we estimate*

$$\hat{\rho}^+ = \frac{1}{m_-}\sum_{i=1}^{m_-}\mathbf{1}_{\tilde{r}_i = 1 \mid r_i^* = 0} \quad \hat{\rho}^- = \frac{1}{m_+}\sum_{i=1}^{m_+}\mathbf{1}_{\tilde{r}_i = 0 \mid r_i^* = 1}.$$

*denoting* $m_{\min} := \min\{m_+, m_-\}$. *Fix a tolerance level* $\delta \in (0, 1)$ *and a failure probability* $\eta \in (0, 1)$. *If*

$$m_{\min} \geq \frac{2}{\delta^2(1 - \rho^+ - \rho^-)^2}\log\frac{4}{\eta},$$

*then with probability at least* $1 - \eta$, $|\lambda - 1| \leq \delta$.

Proposition 5.6 shows that the estimation error concentrates with $m_{\min}$; in our experiments, we therefore reserve a held-out subset and explicitly balance positives/negatives to control it.

This leads to our final practical algorithm, which first estimates the noise rates from data and then uses them to perform a debiased policy update. For simplicity, our experiments estimate a single pair of global flip rates ($\rho^+$, $\rho^-$) shared across prompts; refining this with query-dependent (e.g., prompt-family or difficulty-stratified) rates is left for future work. All these theoretical components are integrated into Algorithm 1 .

## 6. Experiments

We conduct our experiments using two language models: `Qwen3-0.6B` (Yang et al., 2025), `Llama-3.2-1B-Instruct` (Meta, 2024), within the EASYR1 framework (Zheng et al., 2025). Additional synthetic experiments include `Qwen3-1.7B`, `Llama-3.2-3B`, and `Qwen3-4B`. We first validate with synthetic noise where ground-truth rewards are available, then switch to off-the-shelf reward models and estimate flip rates from held-out data. We compare our method against the standard Dr.GRPO and GRPO baselines (with a more focus on Dr.GRPO since it is the corrected version of GRPO (Liu et al., 2025)) by evaluating the clean accuracy $\mathbb{E}_{q \sim \rho_Q}\big[p_{\pi_k}(q)\big]$ measured with the true ground truth reward $r^*$, at each iteration k. A complete description of the experimental setup, including dataset specifics, evaluation protocol, and all hyperparameters, is provided in Appendix B. Our code is available online.[1]

### 6.1. Synthetic noise

To validate our method, we first establish a controlled setup where we have access to ground truth rewards. We synthetically corrupt these rewards by flipping the binary label for each response $(q, o)$ with probability $\rho^+$ (for false positives) and $\rho^-$ (for false negatives), simulating the noisy channel described in Section 4.

To comprehensively assess the robustness of our method under varying conditions, we conduct this experiment across five different pairs of noise rates ($\rho^+$ , $\rho^-$). The aggregated results for GSM8K (Cobbe et al., 2021) , demonstrating the performance of each model with and without our noise correction, are presented in Table 1 and Table 3. Interestingly, for `Llama-3.2-1B` under light noise conditions, the corrected model even outperforms the noiseless baseline. This suggests that the noise-correction mechanism not only mitigates label corruption but can also act as a form of regularization, improving generalization beyond the ideal noiseless case.

To further strengthen our empirical results, we extended Table 1 on Dr.GRPO to include a diverse set of models : `Qwen3-1.7B`, `Llama-3.2-3B` and `Qwen3-4B` on the MATH benchmark (Hendrycks et al., 2021b). Results are presented in Table 2. These additional experiments demonstrate that our findings hold across varying model scales and distinct datasets.

One may also ask whether clipping alone is enough to handle noisy rewards in practical GRPO-style training. We test this in Appendix C with a clipping ablation. The results show that clipping stabilizes updates, but it does not correct

---

[1]Code: `https://github.com/omarito101/GRPO_project`

*Table 1.* Performance on GSM8K under synthetic noise for Dr.GRPO. Final clean accuracy (%).

| LLM | Noise $(\rho^+, \rho^-)$ | Noiseless case | No corr. | With corr. |
|---|---|---|---|---|
| Qwen3-0.6B | (0.1,0.2) | | 69.40 | **71.17** |
| | (0.2,0.3) | | 65.92 | **70.58** |
| | (0.3,0.4) | 74.33 | 62.21 | **68.31** |
| | (0.05,0.15) | | 71.11 | **71.78** |
| | (0.4,0.5) | | 48.38 | **56.34** |
| Llama-3.2-1B | (0.1,0.2) | | 59.84 | **71.04** |
| | (0.2,0.3) | | 52.99 | **68.51** |
| | (0.3,0.4) | 65.69 | 50.18 | **54.66** |
| | (0.05,0.15) | | 60.11 | **70.55** |
| | (0.4,0.5) | | 34.98 | **38.53** |

*Table 2.* Performance on MATH under synthetic noise for Dr.GRPO across additional model scales. Final clean accuracy (%).

| LLM | Noise $(\rho^+, \rho^-)$ | Noiseless case | No corr. | With corr. |
|---|---|---|---|---|
| Qwen3-1.7B | (0.1,0.2) | 48.93 | 39.70 | **43.32** |
| | (0.3,0.4) | | 34.24 | **39.13** |
| Llama-3.2-3B | (0.1,0.2) | 52.76 | 51.01 | **51.70** |
| | (0.3,0.4) | | 52.56 | **53.02** |
| Qwen3-4B | (0.1,0.2) | 52.24 | 39.49 | **45.16** |
| | (0.3,0.4) | | 24.87 | **31.58** |

the systematic bias introduced by false positives and false negatives. In other words, clipping is not a substitute for reward denoising, and our correction remains useful even when clipping is enabled.

The synthetic setting verifies that improvements come from the correction mechanism itself rather than from incidental reward-model quirks. Having validated the mechanism under known flip rates, we now turn to the practical setting: off-the-shelf reward models with unknown noise, where flip rates are estimated from data and correction must work end-to-end. We conduct experiments on two tasks with verifiable ground truth: math reasoning and code generation.

### 6.2. Math tasks

For the purpose of mathematical reasoning, we test our algorithm on GSM8K (Cobbe et al., 2021) using the two pre-trained reward models: `RLHFlow/Llama3.1-8B-ORM-Mistral-Data` and `nvidia/AceMath-7B-RM`.

*Table 3.* Performance on GSM8K under synthetic noise for GRPO. Final clean accuracy (%). (Natarajan corr. only means replacing $\tilde{r}$ by $\hat{r}$ and $M = \sqrt{Var(\hat{r})}$, corr./Z means using $\hat{r}$ with $M = \sqrt{Z}$

| LLM | Noise $(\rho^+, \rho^-)$ | Noiseless case | No corr. | Natarajan corr. only | Natarajan corr./Z |
|---|---|---|---|---|---|
| Qwen3-0.6B | (0.2,0.1) | 78.38 | 75.73 | 75.44 | **77.88** |

Note that another advantage of our method is that it allows us to deploy RL in a setting with limited annotations. We bypass the need for a curated dataset of verified correct answers for every problem, instead leveraging the estimated flip rates of the reward models to provide a scalable, automated feedback signal.

We estimated noise parameters from `Llama3.1-8B-ORM-Mistral-Data` $(\rho^+ = 0.216, \rho^- = 0.324)$ and `AceMath-7B-RM` $(\rho^+ = 0.020, \rho^- = 0.392)$ reward models, where `AceMath-7B-RM` exhibits very low false positive rates but high false negative rates, while `Llama3.1-8B-ORM-Mistral-Data` shows more balanced but generally higher noise levels across both error types. The calibration budget is also consistent with the finite-sample guarantee in Proposition 5.6. GSM8K contains 7,473 training examples, so holding out 20% gives $N_{est} \approx 1,494$ calibration examples. With our balanced calibration construction, this corresponds to $m^+ = m^- = 747$. At confidence level 95%, Proposition 5.6 gives

$$|\hat{\tau} - \tau| \leq \sqrt{\frac{2\log(80)}{747}} \approx 0.108, \qquad \tau = 1 - \rho^+ - \rho^-.$$

For the two reward models above, this corresponds to a relative error bound of approximately 23.5% and 18.4%, respectively. Thus, a 20% calibration split is sufficient to estimate the correction scale at the accuracy level predicted by our theory, while leaving the remaining training examples to be labeled automatically by the reward model.

We apply then our algorithm; the results, presented in Table 4 and Table 5, provide strong empirical evidence for the efficacy of our noise-correction algorithm. We observe a consistent performance improvement across all LLMs when the correction is applied. For instance, `Llama-3.2-1B-Instruct`'s accuracy on GSM8K, when guided by the `AceMath-7B-RM`, substantially increases from 52.52% to 59.21%: a gain of nearly 7 percentage points. Similarly, `Qwen3-0.6B` improves by over 2 points with both reward models. Moreover, Table 5 shows that applying only the mean correction without variance adjustment (Natarajan corr. only) degrades performance, confirming our theoretical prediction that both components are necessary.

These findings validate our approach of modeling reward corruption via estimated flip rates ($\rho^+$ and $\rho^-$); the consistent accuracy gains confirm that our debiasing mechanism successfully counteracts the negative effects of the noisy reward signal, leading to a more effective policy.

### 6.3. Code generation tasks

For code generation evaluation, we test our algorithm on the `APPS` dataset (Hendrycks et al., 2021a) using the pre-trained

*Table 4.* Performance comparison of our algorithm on GSM8K with Dr.GRPO for two LLMs and two reward models.

| LLM | Reward model | Accuracy (%) | |
|---|---|---|---|
| | | w/o Corr | w Corr |
| Qwen3-0.6B | Llama3.1-8B-ORM-Mistral-Data | 72.61 | **74.88** |
| | AceMath-7B-RM | 70.90 | **73.20** |
| Llama-3.2-1B | Llama3.1-8B-ORM-Mistral-Data | 63.98 | **68.50** |
| | AceMath-7B-RM | 52.52 | **59.21** |

*Table 5.* Performance comparison on GSM8K of our algorithm with GRPO for two reward models.

| LLM | Reward model | Accuracy (%) | | |
|---|---|---|---|---|
| | | w/o Corr | Nat. Corr | Nat. Corr/Z |
| Qwen3-0.6B | Llama3.1-8B-ORM-Mistral-Data | 76.47 | 72.82 | **79.12** |
| | AceMath-7B-RM | 72.37 | 73.13 | **75.18** |

reward model: `weqweasdas/RM-Mistral-7B`.

As a remark, our method suggests a path toward more computationally efficient feedback in code generation. Using a corrected reward model could avoid the high cost of sandboxed code execution, though a quantitative analysis of these potential efficiency gains remains a subject for future work.

For code generation tasks, `RM-Mistral-7B` demonstrates extremely high false positive rates ($\rho^+ = 0.53, \rho^- = 0.233$), incorrectly rewarding half of all negative samples.

Results in Table 6 demonstrate that the noise correction method yields consistent, albeit modest, improvements when paired with the highly noisy `RM-Mistral-7B` reward model. For instance, the accuracy of `Qwen3-0.6B` improves from 9.01% to 10.50%, and `Llama-3.2-1B`'s accuracy sees a slight increase from 8.55% to 8.68%. These gains underscore the method's ability to mitigate the impact of the reward model's extremely high false-positive rate.

## 7. Limitations and Future Work

**Scope and future work.** Our analysis focuses on the RLVR setting where each response has a latent binary correctness label $r^*(q, o) \in \{0, 1\}$, and the observed reward is generated by a class-conditional Bernoulli flip channel with false-positive and false-negative rates $\rho^+(q)$ and $\rho^-(q)$. We assume $1 - \rho^+(q) - \rho^-(q) > 0$, so that the observed reward remains positively correlated with the true reward. This setting matches the verifier-based tasks studied in our experiments; for reward-model experiments, continuous scores are thresholded into binary rewards before applying the correction. In practice, we estimate a global pair of flip rates from a small balanced calibration set, allowing the remaining training data to use automatic reward-model feedback.

*Table 6.* Performance comparison of our algorithm on APPS with Dr.GRPO for one reward model.

| LLM | Reward model | Accuracy (%) | |
|---|---|---|---|
| | | w/o Corr | w/ Corr |
| Qwen3-0.6B | RM-Mistral-7B | 9.01 | **10.50** |
| Llama-3.2-1B | RM-Mistral-7B | 8.55 | **8.68** |

Several extensions are natural. Future work can develop query-dependent, difficulty-dependent, or online flip-rate estimation, instead of using global rates. Another promising direction is to design noise-aware regularization schedules motivated by the flattening of the GRPO improvement bound under reward noise. Finally, extending the framework beyond binary RLVR to richer reward channels and evaluating it at larger scale would further clarify the scope of noise-aware policy optimization.

## 8. Conclusion

We analyzed GRPO and Dr.GRPO under noisy binary rewards, as commonly encountered in RLVR when rewards are produced by imperfect verifiers or reward models. We showed that Bernoulli reward corruption systematically attenuates the clean learning signal. In particular, the noisy Dr.GRPO gradient is scaled by the signal-retention factor $1 - \rho^+ - \rho^-$, while GRPO is additionally affected through its variance normalization. Under fixed regularization, this attenuation leads to weaker policy improvement and suboptimal fixed points compared with the clean-reward dynamics.

Motivated by this analysis, we proposed a simple noise-correction method based on estimating the false-positive and false-negative rates of the reward channel. For Dr.GRPO, the correction debiases the reward and recovers the clean population gradient. For GRPO, we showed that reward debiasing alone is not enough, and introduced a variance-adjusted correction to account for the effect of noise on the normalization term.

Empirically, our noise-corrected update successfully improves clean accuracy, yielding gains of up to +6.7 points in math and +1.5 in code across synthetic and realistic settings. While these results validate our theory and provide consistent performance improvements, larger-scale studies involving bigger models and broader domains are required to fully characterize performance in production RLHF/RLVR pipelines.

## Impact Statement

This paper presents work whose goal is to advance the field of Machine Learning. There are many potential societal consequences of our work, none which we feel must be specifically highlighted here.

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

# A. Proofs

**Proof of Theorem 3.2 :** For each policy $\pi$, we define

$$p_\pi(q) \;=\; \mathbb{E}_{o\sim\pi(\cdot|q)}\big[r^*(q,o)\big].$$

and

$$L_{\pi_k}\big(\pi(\cdot\mid q)\big) \;=\; \mathbb{E}_{o\sim\pi(\cdot|q)}\big[A_{\pi_k}(q,o)\big],$$

$$L_{\pi_k}\big(\pi(\cdot\mid q)\big) \;=\; \mathbb{E}_{o\sim\pi(\cdot|q)}\Big[\frac{r^*(q,o) - \mathbb{E}_{o\sim\pi_k(\cdot|q)}[r^*(q,o)]}{\sigma_k(q)}\Big].$$

Writing the following :

$$L_{\pi_k}\big(\pi(\cdot\mid q)\big) - (p_\pi(q) - p_{\pi_k}(q)) = \frac{1-\sigma_k(q)}{\sigma_k(q)}\,\big(p_\pi(q) - p_{\pi_k}(q)\big)$$

And we also have the following using Holder Inequality supposing discrete space for simplicty :

$$\big|p_\pi(q) - p_{\pi_k}(q)\big| = \big|\mathbb{E}_{o\sim\pi(\cdot|q)}\big(r^*(q,o)\big) - \mathbb{E}_{o\sim\pi_k(\cdot|q)}\big(r^*(q,o)\big)\big| \le \|\pi - \pi_k\|_1\,\|r\|_\infty$$

Rewrite with TV, the total variation distance between $P$ and $Q$ two probability measures on a measurable space $(\Omega, \mathcal{F})$, defined as

$$\mathrm{TV}(P,Q) \;=\; \sup_{A\in\mathcal{F}}\big|P(A) - Q(A)\big|.$$

We have then :

$$\big|p_\pi(q) - p_{\pi_k}(q)\big| \le 2\,\mathrm{TV}\big(\pi(\cdot\mid q)\,\|\,\pi_k(\cdot\mid q)\big)\,\|r\|_\infty.$$

And, since $\|r\|_\infty \le 1$ , we then have :

$$p_\pi(q) - p_{\pi_k}(q) \;\le\; 2\left(\frac{1-\sigma_k(q)}{\sigma_k(q)}\right)\mathrm{TV}\big(\pi(\cdot\mid q)\,\|\,\pi_k(\cdot\mid q)\big).$$

Rearrangaring terms in the equation before , we get :

$$p_\pi(q) - p_{\pi_k}(q) \;\ge\; L_{\pi_k}\big(\pi(\cdot\mid q)\big) - 2\left(\frac{1-\sigma_k(q)}{\sigma_k(q)}\right)\mathrm{TV}\big(\pi(\cdot\mid q)\,\|\,\pi_k(\cdot\mid q)\big).$$

Integrating over $q\sim\rho_{\mathcal{Q}}$ :

$$\mathbb{E}_{q\sim\rho_{\mathcal{Q}}}\big[p_\pi(q) - p_{\pi_k}(q)\big] \;\ge\; \mathbb{E}_{q\sim\rho_{\mathcal{Q}}}[L_{\pi_k}\big(\pi(\cdot\mid q)\big)] - 2\,\mathbb{E}_{q\sim\rho_{\mathcal{Q}}}\big[\left(\frac{1-\sigma_k(q)}{\sigma_k(q)}\right)\mathrm{TV}\big(\pi(\cdot\mid q)\,\|\,\pi_k(\cdot\mid q)\big)\big].$$

Applying Cauchy–Schwarz inequality :

$$\mathbb{E}_{q\sim\rho_{\mathcal{Q}}}\big[p_\pi(q) - p_{\pi_k}(q)\big] \;\ge\; \mathbb{E}_{q\sim\rho_{\mathcal{Q}}}[L_{\pi_k}\big(\pi(\cdot\mid q)\big)] - 2\sqrt{\mathbb{E}_{q\sim\rho_{\mathcal{Q}}}\Big[\left(\tfrac{1-\sigma_k(q)}{\sigma_k(q)}\right)^2\Big]}\sqrt{\mathbb{E}_{q\sim\rho_{\mathcal{Q}}}[\mathrm{TV}^2(\pi(\cdot\mid q)\|\pi_k(\cdot\mid q))]}.$$

**Proof of Theorem 3.3 :**   For each $k > 1$, consider the optimization problem :

$$\pi_{k+1} \;=\; \arg\max_{\pi} \; \mathbb{E}_{q\sim\rho_Q}\big[L_{\pi_k}(\pi(\cdot\mid q)) \;-\; \beta\,\mathrm{KL}\big(\pi(\cdot\mid q)\,\|\,\pi_{\mathrm{ref}}(\cdot\mid q)\big)\big]. \tag{11}$$

This problem is concave thanks to the divergence term, and has a closed form solution as shown in the following proposition A.1,

**Proposition A.1** (Closed-Form Update). *The solution of the optimization problem in Equation 11 is the following:*

$$\pi_{k+1}(o\mid q) = \frac{1}{Z_k(q)}\pi_{ref}(o\mid q)\exp\left(\frac{1}{\beta}\left(\omega_\varepsilon^+(p_{\pi_k}(q))\mathbb{1}_{r^*(q,o)=1} - \omega_\varepsilon^-(p_{\pi_k}(q))\mathbb{1}_{r^*(q,o)=0}\right)\right),$$

*where*

$$\omega_\varepsilon^+(p_{\pi_k}(q)) = \sqrt{\frac{1 - p_{\pi_k}(q)}{p_{\pi_k}(q)}}, \quad \omega_\varepsilon^-(p_{\pi_k}(q)) = \sqrt{\frac{p_{\pi_k}(q)}{1 - p_{\pi_k}(q)}},$$

*and*

$$Z_k(q) = p_{ref}(q)\exp\left(\frac{1}{\beta}\omega_\varepsilon^+(p_{\pi_k}(q))\right) + (1 - p_{ref}(q))\exp\left(-\frac{1}{\beta}\omega_\varepsilon^-(p_{\pi_k}(q))\right).$$

$\mathbf{1}_{\{\cdot\}}$ denotes the indicator function, which returns 1 if the condition is true and 0 otherwise.

**Sketch of the proof :**   Using Definition 1, denoting $p_{\pi_k}(q) \;=\; \mathbb{E}_{o\sim\pi_k(\cdot\mid q)}\big(r^*(q,o)\big)$,

$$L_{\pi_k}\big(\pi(\cdot\mid q)\big) \;=\; \mathbb{E}_{o\sim\pi(\cdot\mid q)}\left[\frac{r^*(q,o) - \mathbb{E}_{o\sim\pi_k(\cdot\mid q)}[r^*(q,o)]}{\sigma_k(q)}\right] = \mathbb{E}_{o\sim\pi(\cdot\mid q)}\left[\frac{r^*(q,o) - p_{\pi_k}(q)}{\sqrt{p_{\pi_k}(q)(1 - p_{\pi_k}(q))}}\right].$$

$$L_{\pi_k}\big(\pi(\cdot\mid q)\big) = \underbrace{\sqrt{\frac{1 - p_{\pi_k}(q)}{p_{\pi_k}(q)}}}_{\omega^+(p_{\pi_k})}\mathbb{E}_{o\sim\pi(\cdot\mid q)}[\mathbf{1}_{r^*(q,o)=1}] - \underbrace{\sqrt{\frac{p_{\pi_k}(q)}{1 - p_{\pi_k}(q)}}}_{\omega^-(p_{\pi_k})}\mathbb{E}_{o\sim\pi(\cdot\mid q)}[\mathbf{1}_{r^*(q,o)=0}].$$

The problem is concave so we take the first order conditions supposing the differentiability.

To prove Theorem 3.3 , we use the proposition A.1 and we get :

$$
\begin{aligned}
p_{\pi_k}(q) &= \mathbb{E}_{o\sim\pi_k(\cdot\mid q)}\big(\mathbf{1}_{r(q,o)=1}\big)\\
&= \frac{1}{Z_{k-1}(q)}\int d\pi_{\mathrm{ref}}(o\mid q)\exp\left(\frac{1}{\beta}\big(w^+(p_{\pi_{k-1}})\mathbf{1}_{r^*(q,o)=1} - w^-(p_{\pi_{k-1}})\mathbf{1}_{r^*(q,o)=0}\big)\right)\mathbf{1}_{r(q,o)=1}\\
&= \frac{1}{Z_{k-1}(q)}\exp\left(\frac{1}{\beta}w^+(p_{\pi_{k-1}})\right)\mathbb{E}_{\pi_{\mathrm{ref}}}\big(\mathbf{1}_{r^*(q,o)=1}\big)\\
&= \frac{p_{\mathrm{ref}}(q)\exp\left(\frac{1}{\beta}w^+(p_{\pi_{k-1}})\right)}{Z_{k-1}(q)}\\
p_{\pi_k}(q) &= \frac{1}{1 + \frac{1-p_{\mathrm{ref}}}{p_{\mathrm{ref}}}\exp\left(-\frac{1}{\beta}\big[w^+(p_{\pi_{k-1}}) + w^-(p_{\pi_{k-1}})\big]\right)}\\
\Rightarrow p_{\pi_k}(q) &= h_{\mathrm{pref}}(p_{\pi_{k-1}})\\
\text{where } h_{\mathrm{pref}}(p) &= \frac{1}{1 + \frac{1-p_{\mathrm{ref}}}{p_{\mathrm{ref}}}\exp\left(-\frac{1}{\beta\cdot\sqrt{p(1-p)}}\right)}
\end{aligned}
$$

For stability , we add an $\varepsilon > 0$ in the weights denominator. With this, we end up with the expression of $h_{\varepsilon,\mathrm{pref}}$.

**Proof of Proposition 3.4:** We have :

$$p_{\pi_k}(q) = h_{\text{pref}}(p_{\pi_{k-1}}),$$

where

$$h_{\text{pref}}(p) = \frac{1}{1 + \frac{1 - p_{\text{ref}}}{p_{\text{ref}}} \exp\left(-\frac{1}{\beta\sqrt{p(1-p)}}\right)}.$$

Using :

$$\exp(-\cdot) < 1 \implies 1 + \frac{1 - p_{\text{ref}}}{p_{\text{ref}}} \cdot \exp(-\cdot) < \frac{1}{p_{\text{ref}}}$$

We obtain that : $\quad p_{\pi_k}(q) > p_{\text{ref}}, \; \forall k$

**Proof of Theorem 4.1 :** Using that :

$$\tilde{L}_{\pi_k}(\pi(\cdot \mid q)) = \frac{\mathbb{E}_{\xi, o \sim \pi(\cdot|q)}[\tilde{r}(q, o, \xi)] - \mathbb{E}_{\xi, o \sim \pi_k}[\tilde{r}(q, o, \xi)]}{\sqrt{\text{Var}_{\xi, o \sim \pi_k}[\tilde{r}(q, o, \xi)]}}, \tag{12}$$

$$\tilde{L}_{\pi_k}(\pi(\cdot \mid q)) = \frac{\left(1 - \rho^+(q) - \rho^-(q)\right)\left(\mathbb{E}_{o \sim \pi}[r^*(q, o)] - \mathbb{E}_{o \sim \pi_k}[r^*(q, o)]\right)}{\tilde{\sigma}_k(q)}. \tag{13}$$

The algebraic rearrangement gives this time :

$$\tilde{L}_{\pi_k}\left(\pi(\cdot \mid q)\right) - \left(p_\pi(q) - p_{\pi_k}(q)\right) = \frac{1 - \rho^+(q) - \rho^-(q)) - \tilde{\sigma}_k(q)}{\tilde{\sigma}_k(q)} \left(p_\pi(q) - p_{\pi_k}(q)\right).$$

From here, we apply the same arguments in the proof of Theorem 3.2 to get the final inequality. The only difference is that the term $\frac{1 - \sigma_k(q)}{\sigma_k(q)}$ is replaced by $\frac{1 - \rho^+(q) - \rho^-(q)) - \tilde{\sigma}_k(q)}{\tilde{\sigma}_k(q)}$

**Proof of Theorem 4.2 :** For each $k > 1$, consider the optimization problem :

$$\pi_{k+1} = \arg\max_\pi \mathbb{E}_{q \sim \rho_{\mathcal{Q}}}\left[\tilde{L}_{\pi_k}(\pi(\cdot \mid q)) - \beta\,\text{KL}\left(\pi(\cdot \mid q) \,\|\, \pi_{\text{ref}}(\cdot \mid q)\right)\right]. \tag{14}$$

Developing the equation :

$$\tilde{L}_{\pi_k}(\pi(\cdot \mid q)) = \frac{\left(1 - \rho^+(q) - \rho^-(q)\right)\left(\mathbb{E}_{o \sim \pi}[r^*(q, o)] - \mathbb{E}_{o \sim \pi_k}[r^*(q, o)]\right)}{\tilde{\sigma}_k(q)},$$

We get :

$$\tilde{L}_{\pi_k}(\pi(\cdot \mid q)) = (1 - \rho^+(q) - \rho^-(q))\left[\frac{1 - p_{\pi_k}(q)}{\sqrt{\mu_k(q)(1 - \mu_k(q))}} \mathbb{E}_{o \sim \pi(\cdot|q)}[\mathbb{1}_{r^*(q,o)=1}] - \frac{p_{\pi_k}(q)}{\sqrt{\mu_k(q)(1 - \mu_k(q))}} \mathbb{E}_{o \sim \pi(\cdot|q)}[\mathbb{1}_{r^*(q,o)=0}]\right].$$

The optimization problem is thus :

$$\max_\pi \mathbb{E}_{q \sim \rho_{\mathcal{Q}}}\left[\tilde{w}^+(p_{\pi_k})\,\mathbb{E}_{o \sim \pi(\cdot|q)}\left(\mathbb{1}_{r^*(q,o)=1}\right) - \tilde{w}^-(p_{\pi_k})\,\mathbb{E}_{o \sim \pi(\cdot|q)}\left(\mathbb{1}_{r^*(q,o)=0}\right)\right] - \beta\text{KL}(\pi(\cdot \mid q)\|\pi_{ref}(\cdot \mid q)))$$

where

$$\tilde{w}^+(p_{\pi_k}(q)) = (1 - \rho^+(q) - \rho^-(q))\frac{1 - p_{\pi_k}(q)}{\sqrt{\mu_k(q)(1 - \mu_k(q))}}$$

$$\tilde{w}^-(p_{\pi_k}(q)) = (1 - \rho^+(q) - \rho^-(q))\frac{p_{\pi_k}(q)}{\sqrt{\mu_k(q)(1 - \mu_k(q))}}$$

and

$$\mu_k(q) = \rho^+(q) + (1 - \rho^+(q) - \rho^-(q))p_{\pi_k}(q)$$

From this point, it is exactly the same proof as in the noiseless case replacing $w^+$ and $w^-$ with $\tilde{w}^+$ and $\tilde{w}^-$.

Following the exact same steps , the only thing that will be modified is starting from this line:

$$
\begin{aligned}
p_{\pi_k}(q) &= \frac{1}{1 + \frac{1-p_{\text{ref}}}{p_{\text{ref}}} \exp\left(-\frac{1}{\beta}\left[w^+(p_{\pi_{k-1}}) + w^-(p_{\pi_{k-1}})\right]\right)} \\
&= \frac{1}{1 + \frac{1-p_{\text{ref}}}{p_{\text{ref}}} \exp\left(-\frac{1}{\beta} \cdot \frac{(1-\rho^+(q)-\rho^-(q))}{\sqrt{\mu_{k-1}(q)(1-\mu_{k-1}(q))+\epsilon}}\right)} \\
\Rightarrow p_{\pi_k}(q) &= \tilde{h}_{\varepsilon,\text{pref}}(p_{\pi_{k-1}}) \\
\text{where } \tilde{h}_{\varepsilon,\text{pref}}(p) &= \frac{1}{1 + \frac{1-p_{\text{ref}}}{p_{\text{ref}}} \exp\left(-\frac{1}{\beta} \cdot \frac{(1-\rho^+(q)-\rho^-(q))}{\sqrt{F(p)(1-F(p))+\epsilon}}\right)} \\
\text{with } F(p) &= \rho^+(q) + (1-\rho^+(q)-\rho^-(q))p
\end{aligned}
$$

We observe that, in all cases, what changes in $h$ is the sum $w^+ + w^-$ in the term $\exp\left(-\frac{1 \cdot (\ldots)}{\beta}\right)$.

Noiseless case :

$$w^+ + w^- = \frac{1}{\sqrt{p(1-p)}}$$

Noisy case :

$$w^+ + w^- = \frac{(1 - \rho^+ - \rho^-)}{\sqrt{\mu(p)(1-\mu(p))}} = \frac{(1 - \rho^+ - \rho^-)}{\sqrt{F(p)(1-F(p))}}$$

where

$$F(p) = \rho^+ + (1 - \rho^+ - \rho^-)p$$

Next, to show that $\forall p \in [0,1]$, $\tilde{h}_{\varepsilon,\text{ref}}(p) < h_{\varepsilon,\text{ref}}(p)$, it suffices to prove

$$\frac{1 - \rho^+ - \rho^-}{\sqrt{F(p)(1-F(p)) + \varepsilon}} < \frac{1}{\sqrt{p(1-p) + \varepsilon}}, \tag{15}$$

where $F(p) = \rho^+ + (1 - \rho^+ - \rho^-)p$ and $\varepsilon > 0$.

Since all quantities are strictly positive, we may square and invert while preserving the inequality, obtaining the equivalent condition

$$F(p)(1 - F(p)) + \varepsilon > (1 - \rho^+ - \rho^-)^2\big(p(1-p) + \varepsilon\big). \tag{16}$$

We now expand $F(p)(1 - F(p))$:

$$
\begin{aligned}
F(p)(1 - F(p)) &= \big(\rho^+ + (1 - \rho^+ - \rho^-)p\big)\big(1 - \rho^+ - (1 - \rho^+ - \rho^-)p\big) \\
&= (1 - \rho^+ - \rho^-)^2 p(1-p) + p\,\rho^-(1-\rho^-) + (1-p)\,\rho^+(1-\rho^+). \tag{17}
\end{aligned}
$$

Substituting (17) into (16) and collecting terms yields

$$
\begin{aligned}
&\Big(F(p)(1 - F(p)) + \varepsilon\Big) - (1 - \rho^+ - \rho^-)^2\big(p(1-p) + \varepsilon\big) \\
&= p\,\rho^-(1-\rho^-) + (1-p)\,\rho^+(1-\rho^+) + \big[1 - (1 - \rho^+ - \rho^-)^2\big]\varepsilon. \tag{18}
\end{aligned}
$$

Each term on the right-hand side of (18) is nonnegative, and if $\rho^+ + \rho^- > 0$ then the last term is strictly positive. Therefore (16) holds for all $p \in [0, 1]$, which implies (15).

Consequently,

$$\tilde{h}_{\varepsilon, \text{ref}}(p) < h_{\varepsilon, \text{ref}}(p), \qquad \forall p \in [0, 1].$$

**Proof of Corollary 4.3 :** Since $\tilde{h}_{\varepsilon, \text{ref}}(p) < h_{\varepsilon, \text{ref}}(p)$ for all $p \in [0, 1]$ (from Theorem 4.2), and both functions are continuous and map $[0, 1]$ to itself, their respective fixed points satisfy $p^*_{\text{noisy}} < p^*_{\text{noiseless}}$.

**Proof of Proposition 5.1 :** This follows directly from the definition, the linearity of expectation, and the fact that the expectation of $\tilde{r}$ is given in (4).

**Proof of Theorem 5.2 :** Using :

$$L_{\pi_k}^{\text{Deb}}\big(\pi(\cdot \mid q)\big) = \mathbb{E}_{o\sim\pi(\cdot|q)}\left[\frac{r^*(q, o) - \mathbb{E}_{o\sim\pi_k(\cdot|q)}\big(r^*(q, o)\big)}{M_k(q)}\right].$$

The algebraic rearrangement gives this time

$$L_{\pi_k}^{\text{Deb}}\big(\pi(\cdot \mid q)\big) - \big(p_\pi(q) - p_{\pi_k}(q)\big) = \frac{1 - M_k(q)}{M_k(q)}\big(p_\pi(q) - p_{\pi_k}(q)\big).$$

From here, we apply the same arguments in the proof of Theorem 3.2 to get the final inequality. The only difference is that the term $\frac{1-\sigma_k(q)}{\sigma_k(q)}$ is replaced by $\frac{1-M_k(q)}{M_k(q)}$

**Proof of Proposition 5.3 :**

$$\text{We have :} \quad \mathbb{E}_{\xi\sim U[0,1]}\big[\mathbb{E}_{o\sim\pi_k(\cdot|q)}\big(\hat{r}(q, o, \xi)\big)\big] = \mathbb{E}_{o\sim\pi_k(\cdot|q)}\big(r(q, o)\big) = p_{\pi_k}(q)$$

$$\text{Var}_{\xi,o}\big(\hat{r}(q, o, \xi)\big) = \mathbb{E}_{\xi,o}\big[\hat{r}(q, o, \xi)^2\big] - \big(\mathbb{E}_{\xi,o}\big[\hat{r}(q, o, \xi)\big]\big)^2$$

$$= \frac{\mathbb{E}_{o\sim\pi(\cdot|q)}\big[r^*(q, o)\big]\rho^-(q)(1 - \rho^-(q)) + \big(1 - \mathbb{E}_{o\sim\pi(\cdot|q)}\big[r^*(q, o)\big]\big)\rho^+(q)(1 - \rho^+(q))}{\big(1 - \rho^+(q) - \rho^-(q)\big)^2}$$

$$+ \frac{(1 - \rho^+(q) - \rho^-(q))^2\,\mathbb{E}_{o\sim\pi(\cdot|q)}\big[r^*(q, o)\big]\big(1 - \mathbb{E}_{o\sim\pi(\cdot|q)}\big[r^*(q, o)\big]\big)}{\big(1 - \rho^+(q) - \rho^-(q)\big)^2}$$

$$\text{Var}_{\xi,o}\big(\hat{r^*}(q, o, \xi)\big) = \frac{\mathbb{E}_{o\sim\pi(\cdot|q)}\big[r^*(q, o)\big]\rho^-(q)(1 - \rho^-(q))}{\big(1 - \rho^+(q) - \rho^-(q)\big)^2} + \frac{\big(1 - \mathbb{E}_{o\sim\pi(\cdot|q)}\big[r^*(q, o)\big]\big)\rho^+(q)(1 - \rho^+(q))}{\big(1 - \rho^+(q) - \rho^-(q)\big)^2} + \mathbb{E}_{o\sim\pi(\cdot|q)}\big[r^*(q, o)\big]\big(1 - \mathbb{E}_{o\sim\pi(\cdot|q)}\big[r^*(q, o)\big]\big).$$

Thus :

$$\mathbb{E}_{o\sim\pi(\cdot|q)}\big[r^*(q, o)\big]\big(1 - \mathbb{E}_{o\sim\pi(\cdot|q)}\big[r^*(q, o)\big]\big) = \text{Var}\big(\hat{r^*}(q, o, \xi)\big) - \frac{\mathbb{E}_{o\sim\pi(\cdot|q)}\big[r^*(q, o)\big]\rho^-(q)(1 - \rho^-(q))}{\big(1 - \rho^+(q) - \rho^-(q)\big)^2} - \frac{\big(1 - \mathbb{E}_{o\sim\pi(\cdot|q)}\big[r^*(q, o)\big]\big)\rho^+(q)(1 - \rho^+(q))}{\big(1 - \rho^+(q) - \rho^-(q)\big)^2}$$

Then :

$$\mathbb{E}_{o\sim\pi(\cdot|q)}\big[r^*(q, o)\big]\big(1 - \mathbb{E}_{o\sim\pi(\cdot|q)}\big[r^*(q, o)\big]\big) \approx \widehat{\text{Var}}\big(\hat{r}(q, o, \xi)\big) - \frac{\bar{r}(q)\rho^-(q)(1 - \rho^-(q))}{\big(1 - \rho^+(q) - \rho^-(q)\big)^2} - \frac{\big(1 - \bar{r}(q)\big)\rho^+(q)(1 - \rho^+(q))}{\big(1 - \rho^+(q) - \rho^-(q)\big)^2}.$$

$$\text{where} \quad \bar{r}(q) \equiv \frac{1}{n} \sum_{i=1}^{n} \hat{r}(q, o_i, \xi_i),$$

$$\widehat{\text{Var}}(\hat{r}(q, o, \xi)) \equiv \frac{1}{n-1} \sum_{i=1}^{n} (\hat{r}(q, o_i, \xi_i) - \bar{r}(q))^2.$$

Denote $\quad Z\big((o_i)_{i=1}^n, (\xi_i)_{i=1}^n\big) \equiv \widehat{\text{Var}}(\hat{r}(q, o, \xi)) - \dfrac{\bar{r}(q)\,\rho^-(q)\big(1 - \rho^-(q)\big)}{\big(1 - \rho^+(q) - \rho^-(q)\big)^2} - \dfrac{\big(1 - \bar{r}(q)\big)\,\rho^+(q)\big(1 - \rho^+(q)\big)}{\big(1 - \rho^+(q) - \rho^-(q)\big)^2},$

The estimator is thus unbiased by construction, and :

$$\mathbb{E}_{\xi \sim U[0,1], o \sim \pi(\cdot|q)}\big[Z\big((o_i)_{i=1}^n, (\xi_i)_{i=1}^n\big)\big] = p_{\pi_k}(q) \cdot (1 - p_{\pi_k}(q))$$

**Proof of Theorem 5.5 :** We work with :

$$L_{\pi_k}^{\text{Deb}}\big(\pi(\cdot \mid q)\big) \;=\; \mathbb{E}_{o \sim \pi(\cdot|q)} \mathbb{E}_\xi\left[\frac{\hat{r}(q, o, \xi) - \mathbb{E}_{o \sim \pi_k(\cdot|q)}[\hat{r}(q, o, \xi)]}{M_k(q)}\right].$$

Since we use Natarajan with the estimates of flip rates $\hat{\rho}^+(q)$ and $\hat{\rho}^-(q)$ :

$$L_{\pi_k}^{\text{Deb}}\big(\pi(\cdot \mid q)\big) = \frac{\mathbb{E}_{o \sim \pi(\cdot|q)} \mathbb{E}_\xi\left[\frac{\tilde{r}(q,o,\xi) - \hat{\rho}^+(q)}{1 - \hat{\rho}^+(q) - \hat{\rho}^-(q)}\right] \;-\; \mathbb{E}_{o \sim \pi_k(\cdot|q)} \mathbb{E}_\xi\left[\frac{\tilde{r}(q,o,\xi) - \hat{\rho}^+(q)}{1 - \hat{\rho}^+(q) - \hat{\rho}^-(q)}\right]}{M_k(q)}$$

However :

$$\mathbb{E}_\xi\big(\hat{r}(q, o, \xi)\big) = \rho^+(q) + \big(1 - \rho^+(q) - \rho^-(q)\big)r^*(q, o).$$

Thus

$$L_{\pi_k}^{\text{Deb}}\big(\pi(\cdot \mid q)\big) = \mathbb{E}_{o \sim \pi(\cdot|q)}\left[\frac{1 - \rho^+(q) - \rho^-(q)}{1 - \hat{\rho}^+(q) - \hat{\rho}^-(q)} \cdot \frac{r^*(q, o) - \mathbb{E}_{o \sim \pi_k(\cdot|q)}\big(r^*(q, o)\big)}{M_k(q)}\right].$$

And from that, denoting $M_k'(q) = \frac{1 - \hat{\rho}^+(q) - \hat{\rho}^-(q)}{1 - \rho^+(q) - \rho^-(q)} M_k(q)$, we fall back to the case of Theorem 5.2, and this conclude the proof.

**Proof of Proposition 5.6 :** For any $t_+ > 0$ and $t_- > 0$,

$$\Pr\big(|\hat{\rho}^+ - \rho^+| \ge t_+\big) \le 2\,e^{-2m_- t_+^2}, \qquad \Pr\big(|\hat{\rho}^- - \rho^-| \ge t_-\big) \le 2\,e^{-2m_+ t_-^2}.$$

Indeed , the indicators $\mathbf{1}_{\tilde{r}=1|r^*=0}$ are i.i.d. Bernoulli$(\rho^+)$, and the indicators $\mathbf{1}_{\tilde{r}=0|r^*=1}$ are i.i.d. Bernoulli$(\rho^-)$. Hoeffding's inequality for averages of i.i.d. bounded variables in $[0, 1]$ gives the stated bounds.

Then using union bound, we have for any $t > 0$,

$$\Pr\Big(|(\hat{\rho}^+ + \hat{\rho}^-) - (\rho^+ + \rho^-)| \ge 2t\Big) \;\le\; 2e^{-2m_- t^2} + 2e^{-2m_+ t^2} \;\le\; 4e^{-2m_{\min} t^2}.$$

With $2t = \delta(1 - \rho^+ - \rho^-)$,

$$\Pr\big(|(\hat{\rho}^+ + \hat{\rho}^-) - (\rho^+ + \rho^-)| \ge \delta(1 - \rho_\Sigma)\big) \;\le\; 4\exp\Big(-\tfrac{1}{2}m_{\min}\delta^2(1 - \rho^+ - \rho^-)^2\Big).$$

Requiring the right-hand side to be at most $\eta$ gives the stated lower bound on $m_{\min}$. On this event, $|\lambda - 1| \le \delta$.

**Proof of Theorem 5.4** For any integrable function $f(q, o)$ (with $q$ fixed), we use

$$\nabla_\theta \mathbb{E}_{o\sim\pi_\theta(\cdot|q)}[f(q, o)] = \mathbb{E}_{o\sim\pi_\theta(\cdot|q)}\big[\nabla_\theta \log \pi_\theta(o \mid q) f(q, o)\big],$$

which follows from $\nabla_\theta \pi_\theta(o \mid q) = \pi_\theta(o \mid q)\nabla_\theta \log \pi_\theta(o \mid q)$ and exchanging $\nabla_\theta$ with the integral/sum.

Throughout, we adopt the standard GRPO/Dr.GRPO convention that the quantities computed under $\pi_k$ (e.g. $p_k(q)$, $\sigma_k(q)$, $\mu_k(q)$, $\tilde{\sigma}_k(q)$, and later $M_k(q)$, $Z_k$) are treated as constants with respect to $\theta$ when differentiating.

**Step 1: Gradient of the clean GRPO surrogate.** Recalling the GRPO surrogate at prompt $q$,

$$L_{\pi_k}\big(\pi_\theta(\cdot \mid q)\big) = \mathbb{E}_{o\sim\pi_\theta(\cdot|q)}\left[\frac{r^*(q, o) - p_k(q)}{\sigma_k(q)}\right],$$

we get by the log-derivative identity (and since $p_k(q), \sigma_k(q)$ do not depend on $\theta$):

$$\nabla_\theta L_{\pi_k}\big(\pi_\theta(\cdot \mid q)\big) = \frac{1}{\sigma_k(q)}\mathbb{E}_{o\sim\pi_\theta(\cdot|q)}\Big[\nabla_\theta \log \pi_\theta(o \mid q)\big(r^*(q, o) - p_k(q)\big)\Big].$$

**Step 2: Gradient of the noisy surrogate and comparison factor.** Similarly, the noisy surrogate is

$$\tilde{L}_{\pi_k}\big(\pi_\theta(\cdot \mid q)\big) = \mathbb{E}_{\xi,\, o\sim\pi_\theta(\cdot|q)}\left[\frac{\tilde{r}(q, o, \xi) - \mu_k(q)}{\tilde{\sigma}_k(q)}\right],$$

hence

$$\nabla_\theta \tilde{L}_{\pi_k}\big(\pi_\theta(\cdot \mid q)\big) = \frac{1}{\tilde{\sigma}_k(q)}\mathbb{E}_{\xi,\, o\sim\pi_\theta(\cdot|q)}\Big[\nabla_\theta \log \pi_\theta(o \mid q)\big(\tilde{r}(q, o, \xi) - \mu_k(q)\big)\Big].$$

Using the expectation over the noise gives

$$\nabla_\theta \tilde{L}_{\pi_k}\big(\pi_\theta(\cdot \mid q)\big) = \frac{1 - \rho^+ - \rho^-}{\tilde{\sigma}_k(q)}\mathbb{E}_{o\sim\pi_\theta(\cdot|q)}\Big[\nabla_\theta \log \pi_\theta(o \mid q)\big(r^*(q, o) - p_k(q)\big)\Big].$$

i.e

$$\nabla_\theta \tilde{L}_{\pi_k}\big(\pi_\theta(\cdot \mid q)\big) = \alpha_k(q)\,\nabla_\theta L_{\pi_k}\big(\pi_\theta(\cdot \mid q)\big), \qquad \alpha_k(q) = (1 - \rho^+ - \rho^-)\frac{\sigma_k(q)}{\tilde{\sigma}_k(q)}.$$

**Step 3: Dr.GRPO (centered) comparison.** For the centered Dr.GRPO reward-part surrogate

$$L_{\pi_k}^{\mathrm{Dr}}\big(\pi_\theta(\cdot \mid q)\big) = \mathbb{E}_{o\sim\pi_\theta(\cdot|q)}\big[r^*(q, o) - p_k(q)\big],$$

and its noisy counterpart

$$\tilde{L}_{\pi_k}^{\mathrm{Dr}}\big(\pi_\theta(\cdot \mid q)\big) = \mathbb{E}_{\xi,\, o\sim\pi_\theta(\cdot|q)}\big[\tilde{r}(q, o, \xi) - \mu_k(q)\big],$$

the same argument as Step 2 (without the $\sigma$-normalization) yields

$$\begin{aligned}
\nabla_\theta \tilde{L}_{\pi_k}^{\mathrm{Dr}}\big(\pi_\theta(\cdot \mid q)\big) &= \mathbb{E}_{\xi, o}\Big[\nabla_\theta \log \pi_\theta(o \mid q)\big(\tilde{r} - \mu_k(q)\big)\Big] \\
&= (1 - \rho^+ - \rho^-)\,\mathbb{E}_{o\sim\pi_\theta(\cdot|q)}\Big[\nabla_\theta \log \pi_\theta(o \mid q)\big(r^*(q, o) - p_k(q)\big)\Big] \\
&= (1 - \rho^+ - \rho^-)\,\nabla_\theta L_{\pi_k}^{\mathrm{Dr}}\big(\pi_\theta(\cdot \mid q)\big).
\end{aligned}$$

**Step 4: Debiased gradient and GRPO/Dr.GRPO special cases.** Recall the generalized debiased objective with $\bar{r}(q, o) := \mathbb{E}_\xi[\hat{r}(q, o, \xi)]$:

$$L_{\pi_k}^{\mathrm{Deb}}\big(\pi_\theta(\cdot \mid q)\big) = \mathbb{E}_{o\sim\pi_\theta(\cdot|q)}\left[\frac{\bar{r}(q, o) - \mathbb{E}_{o'\sim\pi_k(\cdot|q)}[\bar{r}(q, o')]}{M_k(q)}\right].$$

By Proposition 5.1, $\bar{r}(q,o) = r^*(q,o)$, hence $\mathbb{E}_{o' \sim \pi_k}[\bar{r}(q,o')] = p_k(q)$ and therefore

$$L_{\pi_k}^{\text{Deb}}\big(\pi_\theta(\cdot \mid q)\big) = \mathbb{E}_{o \sim \pi_\theta(\cdot|q)}\left[\frac{r^*(q,o) - p_k(q)}{M_k(q)}\right].$$

Differentiating (treating $M_k(q)$ as constant w.r.t. $\theta$) gives

$$\nabla_\theta L_{\pi_k}^{\text{Deb}}\big(\pi_\theta(\cdot \mid q)\big) = \frac{1}{M_k(q)}\mathbb{E}_{o \sim \pi_\theta(\cdot|q)}\Big[\nabla_\theta \log \pi_\theta(o \mid q)\,(r^*(q,o) - p_k(q))\Big].$$

*Dr.GRPO case ($M_k = 1$):*
$$\nabla_\theta L_{\pi_k}^{\text{Deb}}\big(\pi_\theta(\cdot \mid q)\big) = \nabla_\theta L_{\pi_k}^{\text{Dr}}\big(\pi_\theta(\cdot \mid q)\big).$$

*GRPO case ($M_k = \sqrt{Z_k}$):* since

$$\nabla_\theta L_{\pi_k}\big(\pi_\theta(\cdot \mid q)\big) = \frac{1}{\sigma_k(q)}\mathbb{E}_{o \sim \pi_\theta(\cdot|q)}\Big[\nabla_\theta \log \pi_\theta(o \mid q)\,(r^*(q,o) - p_k(q))\Big],$$

we obtain the proportionality

$$\nabla_\theta L_{\pi_k}^{\text{Deb}}\big(\pi_\theta(\cdot \mid q)\big) = \frac{\sigma_k(q)}{M_k(q)}\,\nabla_\theta L_{\pi_k}\big(\pi_\theta(\cdot \mid q)\big) = \frac{\sigma_k(q)}{\sqrt{Z_k}}\,\nabla_\theta L_{\pi_k}\big(\pi_\theta(\cdot \mid q)\big),$$

which concludes the proof.

## B. Experiments details

### B.1 - Datasets

We evaluate our methods on two challenging domains: mathematical reasoning and code generation. For mathematical reasoning, we use the **GSM8K** dataset (Cobbe et al., 2021), which contains grade school math word problems requiring multi-step reasoning. For code generation, we use the **APPS** dataset (Hendrycks et al., 2021a) , which contains competitive programming problems of varying difficulty levels.

**GSM8K Processing**: We use the standard train/test split with **7,473 training samples** and **1,319 test samples**, representing an 85/15 split. Input prompts are truncated to 2,048 tokens and responses to 1,024 tokens using the respective model tokenizers.

**APPS Processing**: Following quality filtering to remove samples with empty test cases or missing solutions, we retain **8,215 high-quality samples** (82.2% of the original dataset) with **4,450 training samples** and **3,765 test samples** (54/46 split). We use the official APPS execution framework for ground truth evaluation with 4-second timeout protection.

### B.2 - Models and Implementation

We conduct experiments using two language models of varying scales within the **EasyR1** (Zheng et al., 2025) framework:

- `Qwen3-0.6B-Base` (0.6B parameters) (Yang et al., 2025)
- `Llama-3.2-1B-Instruct` (1B parameters) (Meta, 2024)

All models use their respective chat templates (Llama3.jinja) for proper formatting, with Qwen using the default template.

### B.3 - Reward Models and Noise Estimation

We evaluate four distinct reward configurations representing different types of noisy supervision:

**1. Synthetic Noise**: Applied to ground truth rewards using configurable flip probabilities $\rho^+$ (false positive rate) and $\rho^-$ (false negative rate). We test five noise level pairs: (0.05,0.15), (0.1,0.2), (0.2,0.3), (0.3,0.4), and (0.4,0.5).

**2. AceMath-7B-RM** (NVIDIA): A 7B parameter mathematical reasoning reward model. Noise parameters estimated from confusion matrix analysis: $\rho^+ = 0.020$, $\rho^- = 0.3922$ (very low false positive but high false negative rate).

**3. RLHFlow-Llama3.1-8B-ORM**: An 8B outcome reward model trained on Mistral data. Estimated noise parameters: $\rho^+ = 0.2157$, $\rho^- = 0.3235$ (moderate noise in both directions).

**4. RM-Mistral-7B** (Code): A 7B reward model for code evaluation. Exhibits extremely high noise: $\rho^+ = 0.53$, $\rho^- = 0.233$ (53% false positive rate, representing very noisy supervision).

The reward models used output continuous scores that we threshold: positive scores receive reward 1, negative scores receive 0. This binarization is natural for math and code tasks where answers are objectively correct or incorrect, and allows us to estimate the flip rates $(\rho^+, \rho^-)$ needed for our correction. For code reward models, positive scores indicates functionally correct or high-quality code whereas negative scores indicates buggy or poor-quality implementations.

Noise parameters are estimated using 20% held-out samples from each dataset. We corrupt 50% by introducing mistakes in order to have a balanced dataset for estimation.

### B.4 - Training Configuration

**Optimization**: We use AdamW optimizer with learning rate $1 \times 10^{-6}$, $\beta_1 = 0.9$, $\beta_2 = 0.999$, and weight decay 0.01. Gradient norms are clipped to 1.0.

**Reinforcement Learning**: Training proceeds for 10 epochs with global batch size 64. KL penalty coefficient $\beta = 0.01$ using low-variance KL estimation. Temperature is set to 1.0 during training and 0.5 during validation. We remove the clipping from the algorithm and we work with the sequence level objective function.

**Rollout Configuration**: We test rollout $G = 5$ samples per prompt (used in all comparative experiments)

### B.5 - Evaluation Metrics

**Mathematical Reasoning**: Accuracy computed using exact answer matching with the MathRuler evaluation framework, which handles multiple answer formats (boxed expressions, "The answer is X", etc.).

**Code Generation**: Correctness determined by the official APPS execution framework, testing generated code against provided input/output test cases with proper timeout handling.

**Validation**: Models are evaluated every 5 epochs on the respective test sets. We report final performance after 10 epochs of training.

### B.6 - Reproducibility

All experiments use fixed random seeds (seed=1 for data shuffling, rollout seed=1 for sampling consistency). Configuration files, reward functions, and training scripts are provided in the repository in this link. The framework supports automatic checkpointing every 5 epochs with 3-checkpoint retention for memory efficiency.

**Computational Requirements**: Each experiment runs on a single GPU A100 80 GB. Total computational cost varies by model size and rollout configuration, with typical runtimes of 10-20 hours per experiment depending on dataset size and model parameters.

## C. Clipping Ablation

We ran a 50-step Dr.GRPO clipping ablation on GSM8K under three synthetic noise levels Llama-3.2-1B. We compare four policy-ratio settings: noclip:10000:10000:10000, dapo:0.2:0.28:10.0, ppo:0.2:0.2:3.0, and tight:0.1:0.15:2.0. We can see that clipping does not replace noise correction. In the completed runs, the corrected method remains better in the no-clip, DAPO, and PPO settings across the three noise levels, while very tight clipping largely suppresses the gain and, in the hardest setting, slightly reverses it (33.13 vs 34.36). So the ablation suggests that clipping and correction play different roles: clipping may regularize optimization, but the gains from our method are not explained by clipping alone.

*Table 7.* Clipping ablation on GSM8K with Dr.GRPO for Llama-3.2-1B under synthetic noise. Final clean accuracy (%).

| $(\rho^+, \rho^-)$ | Clip type | w/o corr. | w/ corr. |
|---|---|---|---|
| (0.1, 0.2) | no clip | 51.30 | **58.01** |
| | DAPO clip | 57.40 | **59.64** |
| | PPO clip | 56.95 | **60.95** |
| | tight clip | 50.29 | **51.35** |
| (0.2, 0.3) | no clip | 47.04 | **52.78** |
| | DAPO clip | 52.80 | **57.75** |
| | PPO clip | 53.79 | **58.81** |
| | tight clip | 43.59 | **44.35** |
| (0.3, 0.4) | no clip | 41.77 | **47.40** |
| | DAPO clip | 47.75 | **54.77** |
| | PPO clip | 48.52 | **56.18** |
| | tight clip | **34.36** | 33.13 |

