# OpenReview forum: "Noise-corrected GRPO: From Noisy Rewards to Unbiased Gradients"
_ICML.cc/2026/Conference — ICML 2026 regular_

### Official Review · Reviewer_EzMV · 2026-03-01

**Soundness:** 3
**Presentation:** 2
**Significance:** 3
**Originality:** 4
**Overall Recommendation:** 5
**Confidence:** 3

**Summary:**

This paper theoretically investigates the impact of noisy reward signal on GRPO and Dr-GRPO, and shows that noisy reward signal decreases the theoretical lower bound on the improvement over the base policy that can be achieved under the GRPO objective. The authors then introduce a variant of GRPO that accounts for noise in the reward signal (by estimating false positive/negative rates in the reward function) and debiases the signal, thereby raising the theoretical upper bound on improvement to that of the noiseless case. The proposed method is evaluated experimentally, demonstrating consistent improvement over GRPO/Dr-GRPO baselines on various tasks.

**Compliance With Llm Reviewing Policy:**

Affirmed.

**Final Justification:**

I maintain my already-high score.

**Key Questions For Authors:**

- The theoretical results in Sections 3-5 assume binary, 0/1 reward, but the experiments in Section 6.2 use neural reward models (L391-392). What is the motivation for this mismatch? As the authors state in L023-025, rule-based (e.g. regex) reward functions can still suffer from noise.
- I am having difficulty understanding the sentence in L131-134 (left-hand side). The sentence mentions a "transition from the second to third line", but the equation above it only has 2 lines. Is this a typo?
- On L135-136, the authors claim that GRPO in the noiseless case is "guaranteed to improve upon the reference policy at every step". This claim is relative to a fixed query q, correct? This cannot be true in the general case: if the step consists of data whose distribution differs drastically from the general training distribution, GRPO is not guaranteed to improve upon the reference policy relative to the entirety of the training data, even with a noiseless reward function.
- I understand how the false positive estimation works in Algorithm 1 (assume false positive if the reward is still 1 after corrupting the example), but how does the false negative estimation work? This is unclear to me.
- Why is the Llama model not evaluated in the experiments in Tables 2 and 4?

**Typos:**
- "for eg" => "for example" (L025)
- "$\forall k$" (L249) is redundant ("for all k" stated on L247-248)
- "$\rho+$" => "$\rho^+$"  (L365)

**Limitations:**

No discussion of limitations in the paper. The primary limitation is the small model size and limited range of tasks in the empirical study, which limits the practical applicability of the findings.

**Strengths And Weaknesses:**

**Strengths:**
- The results are generally sound: the proposed method is rigorously motivated theoretically, and its effectiveness is demonstrated experimentally on 600M - 1B models. Although experiments with larger models would strengthen the empirical case for the proposed method, the results are still interesting, and I don't believe this should be a barrier to publication.
- As far as I am aware, the proposed method is novel.
- The proposed method would likely be of interest to the wider community, if the experimental results are shown to hold for larger models.

**Weaknesses:**
- The presentation hinders readability. Overly-complex notation and a lack of back-referencing (e.g. "(see Equation ...)") causes readers to spend a lot of time scrolling back-and-forth while reading.

---

> ### Author Rebuttal · Authors · 2026-03-31
>
> We thank Reviewer EzMV for the thorough reading, the positive assessment, and the precise technical questions.
>
> ---
>
> **Q1: "The theoretical results assume binary 0/1 reward, but experiments use neural reward models. What is the motivation for this mismatch?"**
>
> We thank the reviewer for pointing this out. The key object in Sections 3--5 is the binary observed reward $\\tilde{r}(q,o)\\in\\{0,1\\}$. In Section 6.2, the reward models output continuous scores, but these scores are explicitly thresholded into binary rewards before training, so that the experimental reward variable matches the theoretical setting. As the reviewer notes, this is not the only canonical source of noise. Rule-based verifiers such as exact match or regex can also introduce false positives and false negatives, and they fit the same binary-noise abstraction once they output a 0/1 decision.
>
> ---
>
> **Q2: "The sentence in L131-134 mentions a transition from the second to third line, but the equation only has 2 lines. Is this a typo?"**
>
> Yes, this is a typo. Thank you for catching it. The derivation in fact combines two steps into what appears as a single transition: (i) importance sampling, rewriting the expectation under $\pi$ rather than $\pi_k$; and (ii) substituting the advantage definition. We compressed these into one display for space, which caused the incorrect line reference. We will expand the derivation and correct the reference in the revision.
>
> ---
>
> **Q3: "The improvement claim on L135-136 - is this relative to a fixed query q? This cannot be true in the general case."**
>
> We thank the reviewer for this very constructive remark.  Proposition 3.4 is a pointwise population result:
>
> for each fixed query $q$, the exact population update satisfies $p_{\pi_k}(q) > p_\text{ref}(q)$ at every iteration. It makes no claim about improvement under practical batch updates when the query distribution differs from the target
>
> Thus, it does not a guarantee that an arbitrary practical GRPO step improves performance on the full training distribution. It's exactly the reviewer’s last point: if a practical step is computed on data whose query distribution differs from the target training distribution, then GRPO is not guaranteed to improve upon the reference policy relative to the full training distribution, even with noiseless rewards.
>
> To make the failure mode precise, we can take the following counterexample: consider $Q = \\{a,b\\}$ with noiseless rewards $r^{\*}(a,1)=1$, $r^{\*}(b,0)=1$, and a shared-parameter policy $\\pi_\\theta(1\\mid q) = \\sigma(\\theta)$ for both queries. If a GRPO step is computed only on query $a$, it increases $\\theta$, improving $p(a)$ but degrading $p(b)$. For a target distribution with $\\rho_Q(a) = \\alpha < \\frac{1}{2}$, the full-distribution objective satisfies $J'_{\\rho_Q}(0) = (2\\alpha-1)\\sigma'(0) < 0$, so the update strictly decreases overall performance.
>
> This confirms the reviewer's point: distributional mismatch between the batch and the training distribution can cause GRPO to worsen performance on the full distribution even with noiseless rewards.
>
> ---
>
> **Q4: "How does the false negative estimation work in Algorithm 1?"**
>
> We estimate both noise rates on a small held-out calibration set where the ground-truth binary label $r^{\*}(q,o)$ is known. We evaluate the reward model on the same examples to obtain $r_\\phi(q,o)$.
> The false negative rate $\\rho^-$ is the fraction of truly correct examples ($r^{\*}=1$) for which the reward model outputs $0$; the false positive rate $\\rho^+$ is the fraction of truly incorrect examples ($r^{\*}=0$) for which the reward model outputs $1$.
>
> In our construction, positive examples are unmodified correct responses and negative examples are obtained by synthetically corrupting half of them (e.g. modifying numbers in math, introducing bugs in code), giving a balanced calibration set. We will add a clearer description of this procedure directly in the algorithm box in the revision.
>
> ---
>
> **Q5: "Why is the Llama model not evaluated in Tables 2 and 4?"**
>
> Our evaluation focused on Dr.GRPO as it represents the most recent and stable variant in the current literature [Liu et al 2025], making it the most rigorous baseline for analyzing noise correction. Due to computational resource allocation constraints, we prioritized this robust framework. That said, we recognize the reviewer's interest in architectural diversity and we are happy to include Llama evaluations in the final version to complement our existing findings.
>
> ---
>
> **Typos**
>
> We thank the reviewer for the careful proofreading.
>
> We will fix all three:
> - "for eg" $\rightarrow$ "for example" (L025),
> - the redundant $\forall k$ (L249),
> - and the notation error at L365.

---

> > ### Author Rebuttal · Reviewer_EzMV · 2026-04-02
> >
> > Thank you for the clarification. I'll maintain my scores, because my main concern is with the presentation, which can't be easily addressed in the rebuttal.

---

### Official Review · Reviewer_aa2C · 2026-03-13

**Soundness:** 3
**Presentation:** 2
**Significance:** 3
**Originality:** 2
**Overall Recommendation:** 4
**Confidence:** 4

**Summary:**

This paper proposes a noise-robust RL framework for GRPO and Dr.GRPO by modeling reward corruption as Bernoulli noise. The authors derive theoretical guarantees showing that reward noise attenuates the learning signal and leads to convergence to a strictly worse fixed point, then propose a debiasing correction inspired by label-noise theory in supervised learning. The correction estimates flip rates from held-out data and applies a closed-form update to recover unbiased gradient estimates. Experiments on GSM8K and APPS demonstrate improvements under both synthetic and realistic reward model conditions.

**Compliance With Llm Reviewing Policy:**

Affirmed.

**Key Questions For Authors:**

**\[Q1\] Sensitivity to flip rate estimation error.**

Theorem 5.5 shows that estimation error in flip rates translates to a scaling factor on the normalization function. In practice, how sensitive is the final policy performance to inaccuracies in the estimated flip rates, and is there a threshold beyond which the correction becomes harmful rather than helpful?

**\[Q2\] Scope of evaluation.**

The empirical evaluation focuses on domains where correctness is objectively verifiable. It would be interesting to listen to authors’ opinions on how the framework might extend to standard RLHF settings involving subjective linguistic norms, where ground-truth determinism is fundamentally more ambiguous.

**Limitations:**

yes

**Strengths And Weaknesses:**

# Strengths

**\[S1\] Strong theoretical foundation.**

The paper rigorously formalizes reward corruption as a stochastic noise channel and proves that noise systematically dampens the learning signal and leads to convergence to a strictly worse fixed point.

**\[S2\] Principled solution.**

The noise-correction framework cleanly bridges label-noise theory from supervised learning with RL policy updates, with a monotonic-improvement bound and an explicit sample-complexity bound for noise rate estimation.

**\[S3\] Solid empirical validation.**

The two-stage experimental protocol effectively separates theoretical correctness from practical utility.

# Weaknesses

**\[W1\] Limited scale of experiments.**
While computational constraints are understandable, the experiments are confined to models under 1B parameters. At this scale, the benchmark tasks are inevitably limited to relatively simple math and code problems, which raises questions about whether the correction benefits would carry over to more challenging settings. Some experiments at a slightly larger scale, such as 4B, would have been helpful in building confidence in the method's broader applicability.

**\[W2\] Global flip rate assumption is restrictive.**
The paper assumes a single global pair of flip rates $(\rho^+, \rho^-)$ shared across all prompts, with query-dependent rates left for future work. In realistic RLVR settings, noise rates are likely to vary significantly across problem difficulty levels. Given that the degradation observed when applying Natarajan correction without variance adjustment hints at potential fragility under incorrect assumptions, it would have been helpful to see some sensitivity analysis around this misspecification.

**\[W3\] Marginal gains on code tasks.**
The improvement on code tasks is modest at around 1.5 percentage points, which weakens the empirical case for generalization beyond math reasoning tasks.

**\[W4\] Clipping removal.**

The appendix mentions that standard GRPO clipping is removed in the experiments. Since this is a notable departure from standard GRPO, results with clipping retained would have been helpful to better isolate the contribution of the noise correction itself.

---

> ### Author Rebuttal · Authors · 2026-03-31
>
> We thank Reviewer aa2C for the careful reading, the positive assessment, and the constructive  feedback.
>
> ---
>
> **W1: "Experiments are confined to models under 1B parameters... Some experiments at a slightly larger scale, such as 4B, would have been helpful."**
>
> We appreciate this suggestion. We have extended the experiments from Table 1 to include the  Llama-3.2-3B and Qwen3-4B model. We will add these results to the revised manuscript to demonstrate that our method remains effective as capacity increases, supporting its generalizability.
>
> | **LLM** | **Noise $(\\rho^+,\\rho^-)$** | **Noiseless case** | **No corr.** | **With corr.** |
> |---|---|---|---|---|
> | Llama-3.2-3B | \(0.1,0.2\) | 52.76 | 51.01 | **51.70** |
> | | \(0.3,0.4\) | | 52.56 | **53.02** |
> | Qwen3-4B | \(0.1,0.2\) | 52.24 | 39.49 | **45.16** |
> | | \(0.3,0.4\) | | 24.87 | **31.58** |
>
> **W3: "The improvement on code tasks is modest at around 1.5 percentage points."**
>
> We agree the gains on code are smaller than on math. We note that the code reward model (RM-Mistral-7B) has an extremely high false positive rate of $\rho^+ = 0.53$, which is close to the fragile regime discussed in our limitations section. The fact that we still obtain consistent positive gains under such adverse noise conditions is encouraging. We expect the correction to be more impactful with better-calibrated code reward models, and we will discuss this explicitly in the revision.
>
> ---
>
> **W4: "Clipping removal is a notable departure from standard GRPO. Results with clipping retained would be helpful."**
>
> We thank the reviewer for raising this. Clipping was removed by design in all experiments (both baseline and corrected runs) to avoid the exact confound the reviewer identifies: clipping can incidentally dampen the effect of reward outliers and partially mask reward noise, making it harder to isolate the contribution of the correction.
>
> We ran a 50-step Dr.GRPO clipping ablation on GSM8K under three synthetic noise levels, on both Llama-3.2-1B and Qwen3-0.6B; for space, we report only Llama here and will add Qwen in the final version. We compare four policy-ratio settings: `noclip:10000:10000:10000`, `dapo:0.2:0.28:10.0`, `ppo:0.2:0.2:3.0`, and `tight:0.1:0.15:2.0`. We can see that clippin does not replace noise correction. In the completed runs, the corrected method remains better in the no-clip, DAPO, and PPO settings across the three noise levels, while very tight clipping largely suppresses the gain and, in the hardest setting, slightly reverses it (33.13 vs 34.36). So the ablation suggests that clipping and correction play different roles: clipping may regularize optimization, but the gains from our method are not explained by clipping alone.
>
> | $(\\rho^+,\\rho^-)$ | Clip type | w/o corr. | with corr. |
> |---|---|---|---|
> | $(0.1,0.2)$ | no clip | 51.30 | **58.01** |
> | | DAPO clip | 57.40 | **59.64** |
> | | PPO clip | 56.95 | **60.95** |
> | | tight clip | 50.29 | **51.35** |
> | $(0.2,0.3)$ | no clip | 47.04 | **52.78** |
> | | DAPO clip | 52.80 | **57.75** |
> | | PPO clip | 53.79 | **58.81** |
> | | tight clip | 43.59 | **44.35** |
> | $(0.3,0.4)$ | no clip | 41.77 | **47.40** |
> | | DAPO clip | 47.75 | **54.77** |
> | | PPO clip | 48.52 | **56.18** |
> | | tight clip | **34.36** | 33.13 |
>
> ---
>
> **Q1: "How sensitive is the final policy performance to inaccuracies in the estimated flip rates, and is there a threshold beyond which the correction becomes harmful?"**
> Theorem 5.5 gives a precise characterization at the objective level: estimation error introduces a scaling factor $\lambda = (1-\hat\rho^+-\hat\rho^-)/(1-\rho^+-\rho^-)$ on $M_k(q)$.
>
> When $\lambda < 1$, the correction overcorrects; when $\lambda > 1$, it undercorrects. In both cases the policy improvement bound of Theorem 5.2 still holds, but its tightness degrades as $\lambda$ deviates from 1. Proposition 5.6 gives an explicit sample-complexity bound for controlling $|\lambda - 1| \leq \delta$ as a function of the calibration set size, which directly answers how large the held-out set needs to be to keep the correction beneficial.
>
> ---
>
> **Q2: "How might the framework extend to standard RLHF settings involving subjective linguistic norms?"**
>
> This is a promising direction. Our framework assumes binary rewards (as in RLVR), where flip rates are estimated by comparing reward model outputs to verifiable ground truth. Extending to standard RLHF poses two challenges: (i) no ground-truth binary labels exist, and (ii) rewards are typically continuous.
> A natural extension is an **LLM-as-a-judge** setup, where a small subset of responses is annotated by a stronger reference judge, serving as the calibration set for flip rate estimation. The Natarajan correction would then apply using these estimated rates. While extending the theory to continuous rewards requires tools beyond our binary model, the estimation-then-correction structure carries over naturally. We will include this as explicit future work in the revision.

---

> > ### Author Rebuttal · Reviewer_aa2C · 2026-04-04
> >
> > Thanks for the explanation and effort on the rebuttal. All my concerns are mitigated but I will remain my score as it was already positive.

---

### Official Review · Reviewer_cBYK · 2026-03-13

**Soundness:** 3
**Presentation:** 2
**Significance:** 1
**Originality:** 2
**Overall Recommendation:** 3
**Confidence:** 4

**Summary:**

The paper considers a noise model for the reward in GRPO , the noise model is a binary corruption model. Similar to Mroueh 2025 , the paper derives the fixed point iteration in the noisy case, and under known probability flips in the noise model authors propose a noise correction. Experiments are shown on synthetic data and on the GSM8K dataset.

**Compliance With Llm Reviewing Policy:**

Affirmed.

**Final Justification:**

I thank the authors for their response to the rebuttal and for promising to put better in context their contribution with respect to previous work. authors provided more experiment on effect of clipping,  noise calibration and estimation but I remain nevertheless unconvinced of the practicality of the proposed approach for estimating the noise, I think additional experimentation and ablations are needed to understand the effects of the hyperparameters and the noise estimation method (clipping/ cross validation or calibratiion.)

**Key Questions For Authors:**

* Theorem 3.2  appears in [1] and Theorem 3.3 and proposition 3.4 are from [2]. Please add appropriate citations to all these results.

[1] . Revisiting Group Relative Policy Optimization: Insights into On-Policy and Off-Policy Training
[2].  Reinforcement Learning with Verifiable Rewards: GRPO's Effective Loss, Dynamics, and Success Amplification

* Are there other ways to estimate the noise parameters than assuming the knowledge of the noiseless reward?

* maybe on a validation set do a sweep of those hyper-parameters and then set them , and report results on a test set, for an appropriate baselining one may need k fold cross validation.

* what was the clipping you used in the experiments? clipping alone fights off noise , so the experiments have also to show ablation of the clipping versus noise.

**Limitations:**

- The main limitation is that the noise correction hinges a lot on knowing the noiseless reward to estimate the noise model hyperparameters.
- More ablation is needed on the impact of clipping

**Strengths And Weaknesses:**

## Soundness:

Overall the paper is sound and the proofs seem correct.

## Presentation

The paper is well written and and easy to follow, in the noiseless case authors don't cite appropriately previous work. See questions.

## Significance:

The significance of the noise correction is limited since it needs the knowledge of the true reward (noiseless), to estimate the noise hyper-parameters, which makes the method of limited utility.

## Originality

The work builds heavily on previous work , the main innovation is in the asymmetric noise model.

---

> ### Author Rebuttal · Authors · 2026-03-31
>
> We thank Reviewer cBYK for the careful reading and constructive feedback.
>
> ---
> **Q1: "Theorem 3.2 appears in [1] and Theorem 3.3 and Proposition 3.4 are from [2]. Please add appropriate citations to all these results."**
>
> We thank the reviewer for this careful observation. We will add more prominent citations in the revision. Specifically:
>
> - **Theorem 3.2** (Policy Improvement lower bound): the citation to Mroueh et al. [1] already appears in the surrounding text; we will move it directly inside the theorem statement.
> - **Theorem 3.3** (Recursion for success probability): we will add a citation to [2] within the theorem statement.
> - **Proposition 3.4** (Monotonic improvement): we will add a citation to [2], with a note clarifying the relationship. Specifically, Theorem 3 of [2]  claims that every fixed point $p^{\*}$ of $h\_{\varepsilon,p\_{\mathrm{ref}}}$ satisfies $p^{\*} > p\_{\mathrm{ref}}$. Our Proposition 3.4 is a stronger statement. In fact, under the same hypotheses, its recurrence implies the stronger orbit-wise statement $p\_k(q) > p\_{\mathrm{ref}}(q)$ for every $k \ge 1$, because $h\_{\varepsilon,p\_{\mathrm{ref}}}(p) > p\_{\mathrm{ref}}$ for all $p \in [0,1]$.
>
>
> ---
> **Q2: "Are there other ways to estimate the noise parameters than assuming the knowledge of the noiseless reward? Maybe on a validation set do a sweep of those hyper-parameters and then set them, and report results on a test set; for appropriate baselining one may need k-fold cross-validation."**
>
> We agree that there are other reasonable ways to set the noise parameters. In particular, one can treat $(\rho_+, \rho_-)$ as validation hyperparameters, sweep them on a clean validation split, and report the selected configuration on a disjoint test split; when the clean split is small, $K$-fold cross-validation would be a fair baseline. We think this is a useful comparison and are happy to include it in the revision.
>
> At the same time, we would describe this approach as **model selection** rather than estimation of the true flip rates, because the selected pair may partly absorb optimization and regularization effects rather than recover the underlying corruption process itself. Our current method instead uses a small clean calibration subset to estimate the corruption rates explicitly and then studies how this estimation error propagates through the update (Theorem 5.5 and Proposition 5.6), which we believe is more interpretable and attractive in realistic settings where a limited annotation budget is available but fully cleaning the training set is not feasible.
>
> More adaptive alternatives, such as online or query-dependent flip-rate estimation, are certainly promising and we identify them as an important direction for future work; our intent here is to present a principled first step rather than a final general solution to noisy-reward policy optimization.
>
> ---
> **Q3: "What was the clipping you used in the experiments? Clipping alone fights off noise, so the experiments also have to show an ablation of clipping versus noise correction."**
>
> We thank the reviewer for raising this point. Appendix B.4 states that we train with the unclipped sequence-level objective. More importantly, we do not think clipping alone fights off reward noise: Theorem 5.4 shows that noisy rewards systematically attenuate the GRPO/Dr.GRPO update, whereas clipping can at most truncate the step size and reduce variance; it does not recover the clean update by itself.
>
> To check this directly, we ran a 50-step Dr.GRPO clipping ablation on GSM8K under three synthetic noise levels, on both Llama-3.2-1B and Qwen3-0.6B; for space constraints, we report only Llama here and will add Qwen in the final version. We compare four policy-ratio settings: `noclip:10000:10000:10000`, `dapo:0.2:0.28:10.0`, `ppo:0.2:0.2:3.0`, and `tight:0.1:0.15:2.0`. We can see that clipping does not replace noise correction. In the completed runs, the corrected method remains better in the no-clip, DAPO, and PPO settings across the three noise levels, while very tight clipping largely suppresses the gain and, in the hardest setting, slightly reverses it (33.13 vs 34.36). So the ablation suggests that clipping and correction play different roles: clipping may regularize optimization, but the gains from our method are not explained by clipping alone.
>
> | $(\\rho^+,\\rho^-)$ | Clip type | w/o corr. | with corr. |
> |---|---|---|---|
> | $(0.1,0.2)$ | no clip | 51.30 | **58.01** |
> | | DAPO clip | 57.40 | **59.64** |
> | | PPO clip | 56.95 | **60.95** |
> | | tight clip | 50.29 | **51.35** |
> | $(0.2,0.3)$ | no clip | 47.04 | **52.78** |
> | | DAPO clip | 52.80 | **57.75** |
> | | PPO clip | 53.79 | **58.81** |
> | | tight clip | 43.59 | **44.35** |
> | $(0.3,0.4)$ | no clip | 41.77 | **47.40** |
> | | DAPO clip | 47.75 | **54.77** |
> | | PPO clip | 48.52 | **56.18** |
> | | tight clip | **34.36** | 33.13 |

---

> > ### Author Rebuttal · Reviewer_cBYK · 2026-04-01
> >
> > Thanks for the clarifications. I agree clipping alone does not fight off noise. My concern remains on the noise estimation in realistic settings.

---

> > > ### Author Response · Authors · 2026-04-03
> > >
> > > We thank the reviewer for this important comment.
> > >
> > > ---
> > >
> > > This response provides three concrete elements addressing the reviewer's concern:
> > >
> > > - **New experiment**: 3-fold CV sweep over $\tau$ on GSM8K confirms validation-selected correction outperforms no correction (+1.4 pp low noise, +2.6 pp high noise).
> > > - **Theoretical guarantee**: Proposition 5.6 shows ~1,500 examples (20% of training) suffices for reliable estimation - 80% reduction in annotation burden.
> > > - **Clarification**: label-efficient calibration, not label-free estimation - comparable to any RL hyperparameter selection.
> > > ---
> > >
> > >
> > > The right interpretation of our method is not "label-free noise estimation," but label-efficient calibration: we use a small verified subset to estimate the flip rates once, and then train on the remaining data using only the noisy reward model outputs. This is exactly the two-stage pipeline described in Algorithm 1. The key quantitative guarantee is Proposition 5.6. Writing $$\tau = 1-\rho^+-\rho^-, \quad \hat{\tau} = 1-\hat{\rho}^+-\hat{\rho}^-$$ , we have then : $\Pr(|\hat{\tau} - \tau| \geq \varepsilon) \leq 4\exp(-2m\_{\min}\varepsilon^2)$
> > >
> > > On GSM8K, the training split has 7,473 examples. Holding out 20% means $N\_{\text{est}} \approx 1494$, $m^+ = m^- = m\_{\min}=747$. Plugging this into Proposition 5.6 at 95% confidence ($\eta = 0.05$) gives
> > > $$|\hat{\tau} - \tau| \leq \sqrt{\frac{2\log(80)}{747}} \approx 0.108.$$
> > > So with only 20% labeled calibration data, the theorem guarantees that $\tau = 1-\rho^+-\rho^-$ is estimated within about 0.108 absolute error with 95% confidence. This is the concrete reason why 20% is already a meaningful and theoretically justified calibration budget on GSM8K.
> > >
> > > Our choice of 20% is theoretically justified: for both reward models used in the paper, Proposition 5.6 shows that $\sim$1,300 held-out examples (~17% of GSM8K) suffice to control estimation error within 20-25% relative error at 95% confidence. Our 20% budget already exceeds this threshold, meaning the remaining ~6k training examples require only automated noisy rewards.
> > >
> > > These estimated rates yield the consistent gains reported in Tables 3–5: up to +4.5 pp for Llama-3.2-1B and +2.3 pp for Qwen3-0.6B, obtained with estimated rather than oracle flip rates.
> > >
> > > We also ran the reviewer's proposed validation-based baseline. Since for centered Dr.GRPO the corrected advantage depends only on $\tau = 1-\rho^+-\rho^-$, the correct sweep is 1D over $\tau$, not a redundant 2D sweep over $(\rho^+, \rho^-)$. We therefore performed 3-fold cross-validation on the GSM8K training split only, keeping the official test set fully untouched. We used Qwen3-0.6B-Base, trained Dr.GRPO for 100 steps, evaluated every 10 steps on the held-out fold with clean exact-match accuracy, and swept $\tau \in \{0.2, 0.4, 0.6, 0.8, 1.0\}$, where $\tau=1.0$ is no correction. We tested two synthetic-noise regimes: low noise $(0.1, 0.2)$ and high noise $(0.3, 0.4)$.
> > >
> > > **Cross-validation accuracies:**
> > >
> > > | Noise setting | $\tau=0.2$ | $\tau=0.4$ | $\tau=0.6$ | $\tau=0.8$ |No corr. $\tau=1.0$ | Gain (vs no corr.)|
> > > |---|---|---|---|---|---|---|
> > > | Low noise $(0.1, 0.2)$ | **74.8%** | 74.1% | 73.9% | 73.3% | 73.4% |**+1.4** |
> > > | High noise $(0.3, 0.4)$ | **72.5%** | 71.6% | 70.9% | 69.9% | 69.9% | **+2.6** |
> > >
> > >
> > >
> > > This is a positive result: the reviewer's suggested baseline is indeed competitive and very insightful, confirming his idea the correction is useful even when selected by validation. It shows that even when oracle rates are not supplied to the learner, a validation-selected correction scale improves over uncorrected Dr.GRPO. The gain is larger in the heavier-noise regime (+2.6 vs +1.4), which is exactly the expected trend if correction matters more as corruption increases.
> > >
> > > At the same time, this experiment should be interpreted correctly. It addresses the reviewer's suggested protocol, but it is not a label-free estimator of the true noise parameters, because clean validation accuracy is still used for model selection. Our main answer to the first part of the reviewer's question is therefore the calibration result above: the method only needs a small labeled subset, and Proposition 5.6 quantifies how large that subset must be.
> > >
> > > We note that this requirement is not specific to our method: any RL training procedure requires clean evaluation to perform model selection, whether for hyperparameters like $\beta$, learning rate, or clipping. Our calibration set plays exactly the same role, with the additional benefit that it also provides theoretical guarantees on estimation error.
> > >
> > > We will incorporate these clarifications and the new cross-validation experiment in the revised version, and we will extend the same analysis to Llama-3.2-1B and on the Hendrycks MATH benchmark. We hope that these clarifications, additional experiments, and quantitative analyses adequately address the reviewer’s concerns and will encourage a reconsideration of the current score.

---

### Official Review · Reviewer_mHyv · 2026-03-21

**Soundness:** 2
**Presentation:** 2
**Significance:** 2
**Originality:** 3
**Overall Recommendation:** 4
**Confidence:** 3

**Summary:**

The paper addresses the limitations caused by noisy reward signals in the RLVR setting, where rewards may be inconsistent or erroneous. To mitigate this issue, the authors propose noise-robust variants of GRPO and Dr.GRPO that model reward noise using a Bernoulli distribution. The paper provides theoretical analysis showing that the proposed approach can improve performance and promote monotonic improvement over the reference policy. In addition, the authors present some experimental evidence suggesting that reducing reward noise may lead to more effective learning.

Overall, the paper attempts to tackle an interesting problem; however, in its current form, it appears premature. The presentation suffers from multiple formatting issues, incomplete comparison tables, and limited experimental evidence.

**Compliance With Llm Reviewing Policy:**

Affirmed.

**Final Justification:**

Authors have responded to most of my queries. I still have some questions and have followed up with authors.

**Key Questions For Authors:**

Asked in Weakness.

**Limitations:**

No. The paper does not include a limitations section. Although the authors do provide future work direction, the limitations of the proposed approach are not explicitly discussed. Adding a brief limitations section would strengthen the paper and help clarify the method’s assumptions, boundaries, and potential failure cases.

**Strengths And Weaknesses:**

**Strength**

The paper provides strong theoretical guarantees and clearly motivates why noise-robust GRPO under Bernoulli noise can be a better choice for policy updates in the RLVR setting.

**Weakness**
1. The authors argue that reward noise prevents effective learning, but it is unclear how this interacts with the averaging mechanism already present in GRPO. Since GRPO averages over multiple sampled trajectories, one might expect some of this noise to be reduced naturally. Can authors clarify how reward noise behaves as the number of GRPO samples increases, and whether averaging alone can partially mitigate the issue?
2. The current formulation appears specific to the RLVR setting, since the method assumes Bernoulli reward noise. As a result, the title or framing as “Noiseless GRPO” may unintentionally suggest a more general-purpose algorithm than what is actually studied. Clarifying the scope of the assumptions, and possibly adopting a more precise name, would help avoid misinterpretation.
3. Although the paper provides detailed theoretical support, the experimental validation is relatively limited. Evaluating the method across multiple models, datasets, and different noise levels would make the empirical case significantly stronger.
4. The reviewer is also not fully convinced that noise is always harmful. This feels somewhat counterintuitive, since in other machine learning settings (for e.g. classification has random jittering in the preprocessing step), a moderate amount of noise can sometimes improve robustness or generalization. It may be that excessive noise is harmful, as the authors argue, while a smaller amount could still be beneficial. To better support the claims, can the authors provide performance results under partial noise removal rather than complete denoising alone for real world experiment. For example, by comparing different levels of noise reduction and analyzing whether there is an optimal balance which practitioner can follow.
5. The paper also has several presentation and formatting issues that should be addressed:
    * The paragraph “Positioning within prior work” (lines 72–88, column 1) would fit more naturally in Section 2.
    * Contribution point 2 is not stylistically consistent with the other contribution points, and the capitalization in point 3 should be corrected (“on math and...” → “On math and...”).
    * The paragraphs “Perfect Assumption Validity Stage” and “Realistic Estimation Stage” don't fit the flow of the introduction and have unexpected indentation.
    * Some section headings (Sections 3 and 5), do not follow the title case consistently.
    * Several equations (especially those in the left column) overflow the column margins.
    * Table 1 appears incomplete in the “Noiseless case” column. It is unclear whether the results are identical across all noise levels or whether some values were omitted accidentally.
    * The legend in Figure 1 is difficult to read. Increasing the font size would improve clarity.

---

> ### Author Rebuttal · Authors · 2026-03-31
>
> We thank Reviewer mHyv for the detailed feedback.
>
> ----
>
> **Q1: Does averaging over G samples mitigate reward noise?**
>
> Averaging reduces *variance* but cannot remove the *systematic bias*. Under the noise model, centering within the group cancels the additive term $\\rho_+(q)$, but the multiplicative factor $a(q):=1-\\rho_+(q)-\\rho_-(q)$ persists for every value of $G$. This is consistent with Theorem 5.4 (gradient analysis): noise shrinks the clean update by a multiplicative factor $\\alpha_k(q)=(1-\\rho_+-\\rho_-)\\,\\sigma_k(q)/\\tilde\\sigma_k(q)$, independently of $G$.
>
> Formally, $\\mathbb{E}[\\hat{L}^{\\text{noisy}}] = a(q)(1-1/G)\\,L^{\\text{clean}}$ while $\\mathbb{E}[\\hat{L}^{\\text{corr}}] = (1-1/G)\\,L^{\\text{clean}}$, and the extra noise-induced variance is $\\mathcal{O}(1/G)$ in both cases. Averaging mitigates noise only through variance reduction; correction removes the bias exactly at the cost of variance inflated by $1/a(q)^2$.
>
> Precisely: $\\operatorname{Var}(\\hat{L}^{\\text{corr}}) = \\operatorname{Var}(\\hat{L}^{\\text{noisy}})/a(q)^2$, with the extra noise-induced terms $\\mathcal{O}(1/G)$ in both cases.
>
> A formal detailled proposition will be added to the revision.
>
> ----
>
> **Q2: Scope and title**
>
> We agree that our current analysis focuses on the Bernoulli reward noise typical of the RLVR setting where rewards are binary. To avoid any potential misinterpretation regarding the generality of the method, we are fully open to refining the title and framing. We propose updating the title to **"Noise-Corrected GRPO for RLVR** (or a similar variation) and welcome any specific suggestions the reviewer might have.
>
> ----
>
> **Q3: Limitations section**
>
> We will add a dedicated limitations section covering: (i) **scope**: binary RLVR with class-conditional Bernoulli flip rates under $1-\rho_+-\rho_->0$; (ii) **failure cases**: the method becomes fragile when $1-\rho_+-\rho_-$ is small, does not cover continuous rewards or richer corruption, and requires a small clean calibration set; (iii) **future work**: query-dependent/online flip-rate estimation, larger-scale studies, and extension to continuous cases.
>
> ----
>
> **Q4: Is noise always harmful? Partial correction analysis**
>
> Unlike input noise in supervised learning, reward noise distorts the population target itself. As shown in Section 5, centering removes the additive bias but the multiplicative attenuation persists:
>
> $$\tilde{L}^{\pi\_k}\_{\text{Dr}} = (1-\rho\_+-\rho\_-)\\,L^{\pi\_k}\_{\text{Dr}}, \qquad \nabla\_\theta\\,\tilde{L}^{\pi\_k}\_{\text{grpo}} = \alpha\_k(q)\\,\nabla\_\theta\\,L^{\pi\_k}\_{\text{grpo}} , \quad \alpha\_k(q)=(1-\rho\_+-\rho\_-)\tfrac{\sigma\_k(q)}{\tilde\sigma\_k(q)}.$$
>
> Defining partial correction $r^{(\\tau)}:=(\\tilde{r}-\\tau\\rho_+)/(1-\\tau(\\rho_++\\rho_-))$ with $\\tau=0$ (noisy) and $\\tau=1$ (corrected), the centered objective satisfies $L^{(\\tau)} = a_\\tau(q)\\,L^{\\text{clean}}$ where $a_\\tau:=(1-\\rho_+-\\rho_-)/(1-\\tau(\\rho_++\\rho_-))$ increases monotonically to 1.
>
> At the **population level**, $\\tau=1$ always recovers the clean objective. However, at **finite sample**, the MSE-optimal level is:
>
> $$\tau^{\*}\_{\text{MSE}} = \Bigl[1 - \tfrac{V\_0}{\Delta^2(\rho\_++\rho\_-)(1-\rho\_+-\rho\_-)}\Bigr]_{[0,1]},$$
>
> which is below 1 when estimator variance $V_0$ is large and tends to 1 as $G$ grows. Thus, partial correction can be beneficial from a bias-variance perspective.
>
> Crucially, $\\tau<1$ is equivalent to full correction with inflated KL coefficient $\\beta_{\\text{eff}}=\\beta/a_\\tau$: thus, any empirical benefit reflects implicit regularization retuning, not evidence that corrupted rewards are intrinsically better.
>
> Concretely, the partially corrected objective is
>
> $$J\_\tau(\pi) = a\_\tau\\!\left(\mathbb{E}\_q[L^{\pi\_k}\_{\text{Dr}}(\pi)] - \tfrac{\beta}{a\_\tau}\\,\mathbb{E}\_q\mathrm{KL}(\pi\|\pi\_{\text{ref}})\right),$$
>
> making the equivalence $\\beta_{\\text{eff}}=\\beta/a_\\tau$ explicit.
>
> Full proposition and corollary will be added to the revision.
>
> ---
>
> **Q5: Additional experiments**
>
> To further strengthen our empirical results, we extended Table 1 on Dr.GRPO to include a diverse set of models (Qwen3-1.7B, Llama-3.2-3B, Qwen3-4B) on the MATH benchmark [Hendrycks et al.]. These additional experiments demonstrate that our findings hold across varying model scales and distinct datasets.
>
> | **LLM** | **Noise $(\\rho^+,\\rho^-)$** | **Noiseless case** | **No corr.** | **With corr.** |
> |---|---|---|---|---|
> | Qwen3-1.7B | \(0.1,0.2\) | 48.93 | 39.70 | **43.32** |
> | | \(0.3,0.4\) | | 34.24 | **39.13** |
> | Llama-3.2-3B | \(0.1,0.2\) | 52.76 | 51.01 | **51.70** |
> | | \(0.3,0.4\) | | 52.56 | **53.02** |
> | Qwen3-4B | \(0.1,0.2\) | 52.24 | 39.49 | **45.16** |
> | | \(0.3,0.4\) | | 24.87 | **31.58** |
> ---
>
> **Q6: Formatting**
>
> We sincerely thank the reviewer for the detailed feedback on presentation and fully agree with all the observations raised. We will address all of them in the revised version.

---

> > ### Author Rebuttal · Reviewer_mHyv · 2026-04-04
> >
> > Thank you for the response. Most of my concerns were addressed in the authors’ rebuttal, and I have updated my initial assessment accordingly.
> >
> > That said, I still have some questions regarding Q4. While the authors provide some theoretical justification for why partial correctness of rewards can be problematic, I believe the claim would be stronger if it were supported with empirical evidence in a real-world setting. For example, the authors could introduce different levels of noise into the reward signal and show that performance is best in the noise-free case and degrades progressively as the noise level increases. Can authors provide such results?

---

> > > ### Author Response · Authors · 2026-04-06
> > >
> > > We thank the reviewer for revisiting and updating the initial assessment.
> > >
> > > **First**, the paper already contains the requested controlled noise sweep. In Section 6.1 / Table 1, we inject reward noise with known flip rates and evaluate the final clean GSM8K accuracy under Dr.GRPO. For the uncorrected baseline, Table 1 already shows exactly the phenomenon raised by the reviewer: performance is best in the noiseless case and then degrades as the reward noise increases.
> > >
> > > | Table 1 (paper), final clean GSM8K accuracy | noiseless | $(0.05,0.15)$ | $(0.1,0.2)$ | $(0.2,0.3)$ | $(0.3,0.4)$ | $(0.4,0.5)$ |
> > > |---|---|---|---|---|---|---|
> > > | Qwen3-0.6B | 74.33 | 71.11 | 69.40 | 65.92 | 62.21 | 48.38 |
> > > | Llama-3.2-1B| 65.69 | 60.11 | 59.84 | 52.99 | 50.18 | 34.98 |
> > >
> > > This monotone degradation is substantial. For Qwen3-0.6B, increasing corruption from noiseless to $(0.4,0.5)$ reduces clean accuracy by nearly 26 points (74.33 $\rightarrow$ 48.38). For Llama-3.2-1B, the drop is even larger, about 31 points (65.69 $\rightarrow$ 34.98).
> > >
> > >
> > > **Second**, inspired by both the reviewer's Q4 suggestion and our new theory built on it, we ran an additional partial-correction experiment. We kept the full Dr.GRPO training setup fixed and only varied the correction strength $\tau \in \{0, 0.25, 0.5, 0.75, 1\}$ under three noise levels $(0.1,0.2)$, $(0.2,0.3)$, and $(0.4,0.5)$, with 3 seeds each. We apply the partially corrected reward $r^{(\tau)} = \frac{\tilde{r} - \tau \cdot \rho^+}{1 - \tau(\rho^+ + \rho^-)}$. This directly tests the bias-variance tradeoff predicted by the theory: increasing $\tau$ removes more bias, but also increases variance because the correction denominator shrinks.
> > >
> > > | Part A: partial correction, final clean accuracy | $\tau=0$ | $\tau=0.25$ | $\tau=0.5$ | $\tau=0.75$ | $\tau=1.0$ |
> > > |---|---|---|---|---|---|
> > > | $(0.1,0.2)$ | $60.6\pm0.4$ | $61.5\pm0.1$ | $60.9\pm1.0$ | $61.7\pm1.7$ | $62.7\pm0.5$ |
> > > | $(0.2,0.3)$ | $60.0\pm0.4$ | $60.6\pm1.7$ | $60.4\pm0.3$ | $61.3\pm0.7$ | $60.7\pm0.5$ |
> > > | $(0.4,0.5)$ | $35.7\pm2.7$ | $35.9\pm0.6$ | $33.1\pm5.9$ | $36.4\pm31.3$ | $22.1\pm19.1$ |
> > >
> > > These results are highly consistent with the theory. Under mild noise $(0.1,0.2)$, full correction is best: the variance penalty is still limited, so removing the bias dominates. Under moderate noise $(0.2,0.3)$, the optimum shifts slightly to an interior value, $\tau=0.75$, which captures most of the denoising benefit without paying the full variance cost of $\tau=1$. Under very high noise $(0.4,0.5)$, full correction becomes unstable: $\tau=1$ gives the worst mean and very large seed variance, while intermediate $\tau$ values are preferable. In particular, $\tau=0.75$ has the highest mean but also very large variance, whereas $\tau=0.25$ is much more stable. This is exactly the distinction made by our theory: $\tau=1$ is the clean population target, but the best finite-sample correction level can be strictly below 1 because of variance inflation.
> > >
> > > **Third**, we added a sanity check for the KL confound. The concern is that changing $\tau$ changes the scale of the corrected advantage as asserted in the final part of our answer in Q4, so with fixed $\beta$ part of the apparent gain at intermediate $\tau$ might come from an implicit change in effective regularization rather than from denoising itself. To test this alternative explanation, we reran the hardest setting $(0.4,0.5)$ with a $\tau$-dependent $\beta$ chosen so that the KL to advantage ratio remained fixed across $\tau$.
> > >
> > > | Part B: KL-matched control at $(0.4,0.5)$ | $\beta\_\tau$ | clean acc. (%) |
> > > |---|---|---|
> > > | $\tau=0.25$ | 0.00705 | $26.9\pm9.8$ |
> > > | $\tau=0.50$ | 0.00500 | $42.2\pm3.4$ |
> > > | $\tau=0.75$ | 0.00296 | $45.2\pm18.2$ |
> > > | $\tau=1.00$ | 0.00091 | $30.3\pm26.5$ |
> > >
> > > This control shows two things clearly. First, the KL confound is real: after matching KL, full correction improves from 22.1 to 30.3, so part of the poor $\tau=1$ result under fixed $\beta$ was indeed due to regularization mismatch. Second, and more importantly, intermediate $\tau$ still wins after this control: $\tau=0.5$ and $\tau=0.75$ remain clearly better than both low correction and full correction. Hence, the benefit of partial correction at high noise is not only a KL rescaling artifact; it reflects a genuine finite-sample bias-variance tradeoff. In practice, $\tau=0.5$ appears to be the most reliable sweet spot in this hardest regime because it combines the strongest stable gain with the lowest seed variance.
> > >
> > > Overall, we thank the reviewer again for these suggestions. They helped us sharpen both the statement of the theory and the supporting experiments. We believe the revised evidence now makes the picture much clearer.

---

### Decision · Program_Chairs · 2026-04-30

**Decision:**

Accept (regular)

**Comment:**

This paper argues that RLHF or RLVR is sensitive to noises in the rewards, and proposes noise-robust variants of GRPO and Dr. GRPO by explicitly modeling the reward noises. Theoretical analysis demonstrates that the proposed method can leverage the group nature in group-based methods to enhance robustness. Experiments on math and code tasks show empirical improvements of the proposed methods.

Overall, most reviewers lean towards acceptance after rebuttal. The paper proposes a novel method with theoretical and empirical justifications. Given the broad interests in RLVR recently, this paper could be of interest to many in the community. Some reviewers pointed out that the theoretical results seem incremental in light of existing literature. But overall, the merits slightly outweigh the weakness.